# Fantastic Generalization Measures and Where to Find Them

**Yiding Jiang**[*†], **Behnam Neyshabur**[*], **Hossein Mobahi, Dilip Krishnan, Samy Bengio**
Google Research

{ydjiang,neyshabur,hmobahi,dilipkay,bengio}@google.com

## Abstract

Generalization of deep networks has lately been of great interest, resulting in a number of theoretically and empirically motivated complexity measures. However, most papers proposing such measures study only a small set of models, leaving open the question of whether the conclusion drawn from those experiments would remain valid in other settings. We present the first large scale study of generalization in deep networks. We investigate more then 40 complexity measures taken from both theoretical bounds and empirical studies. We train over 10,000 convolutional networks by systematically varying commonly used hyperparameters. Hoping to uncover potentially causal relationships between each measure and generalization, we analyze carefully controlled experiments and show surprising failures of some measures as well as promising measures for further research.

## 1 Introduction

Deep neural networks have seen tremendous success in a number of applications, but why (and how well) these models generalize is still a mystery (Neyshabur et al., 2014; Zhang et al., 2016; Recht et al., 2019). It is crucial to better understand the reason behind the generalization of modern deep learning models; such an understanding has multiple benefits, including providing guarantees for safety-critical scenarios and the design of better models. A number of papers have attempted to understand the generalization phenomenon in deep models from a theoretical perspective e.g. (Neyshabur et al., 2015b; Bartlett et al., 2017; Neyshabur et al., 2018a; Golowich et al., 2017; Arora et al., 2018; Nagarajan and Kolter, 2019a; Wei and Ma, 2019a; Long and Sedghi, 2019). The most direct and principled approach for studying generalization in deep learning is to prove a *generalization bound*; typically an upper bound on the test error based on some quantity that can be calculated on the training set. Unfortunately, finding tight bounds has proven to be an arduous undertaking. While encouragingly Dziugaite and Roy (2017) showed that PAC-Bayesian bounds can be optimized to achieve a reasonably tight generalization bound, current bounds are still not tight enough to accurately capture the generalization behavior. Others have proposed more direct empirical ways to characterize generalization of deep networks without attempting to deriving bounds (Keskar et al., 2016; Liang et al., 2017). However, as pointed by Dziugaite and Roy (2017), empirical correlation does not necessarily translate to a casual relationship between a measure and generalization.

A core component in (theoretical or empirical) analysis of generalization is the notion of *complexity measure*; a quantity that monotonically relates to some aspect of generalization. More specifically, lower complexity should often imply smaller generalization gap. A complexity measure may depend on the properties of the trained model, optimizer, and possibly training data, but should not have access to a validation set. Theoretically motivated complexity measures such as VC-dimension, norm of parameters, etc., are often featured as the major components of generalization bounds, where the monotonic relationship between the measures and generalization is mathematically established. In contrast, empirically motivated complexity measures such as sharpness (Keskar et al., 2016) are justified by experimentation and observation. In this work, we do not need to distinguish between theoretically vs empirically motivated measures, and simply refer to both as complexity measures.

---

[*]Contributed equally.
[†]Work done as part of the Google AI Residency Program.

Despite the prominent role of complexity measures in studying generalization, the empirical evaluation of these measures is usually limited to a few models, often on toy problems. A measure can only be considered reliable as a predictor of generalization gap if it is tested extensively on many models at a realistic problem size. To this end, we carefully selected a wide range of complexity measures from the literature. Some of the measures are motivated by generalization bounds such as those related to VC-dimension, norm or margin based bounds, and PAC-Bayesian bounds. We further selected a variety of empirical measures such as sharpness (Keskar et al., 2016), Fisher-Rao norm (Liang et al., 2017) and path norms (Neyshabur et al., 2017).

In this study, we trained more than 10,000 models over two image classification datasets, namely, CIFAR-10 (Krizhevsky et al., 2014) and Street View House Numbers (SVHN) Netzer et al. (2011). In order to create a wide range of generalization behaviors, we carefully varied hyperparameters that are believed to influence generalization. We also selected multiple optimization algorithms and looked at different stopping criteria for training convergence. Details of all our measures and hyperparameter selections are provided in Appendix D. Training under all combination of hyperparameters and optimization resulted in a large pool of models. For any such model, we considered ***40 complexity measures***. The key findings that arise from our large scale study are summarized below:

1. It is easy for some complexity measures to capture spurious correlations that do not reflect causal insights about generalization; to mitigate that we propose a more rigorous method to study them.
2. Many norm-based measures not only perform poorly, but *negatively* correlate with generalization specifically when the optimization procedure injects some stochasticity. In particular, the generalization bound based on the product of spectral norms of the layers (similar to that of Bartlett et al. (2017)) has very strong negative correlation with generalization.
3. Sharpness-based measures like PAC-Bayesian bounds (McAllester, 1999) and the measure proposed by Keskar et al. (2016) perform best and seem promising candidates for further research.
4. Measures related to the optimization procedures such as the gradient noise and the speed of the optimization can be predictive of generalization.

Our findings on the relative success of sharpness-based and optimization-based complexity measures for predicting the generalization gap can provoke further study of these measures.

## 1.1 RELATED WORK

The theoretically motivated measures that we consider in this work belong to a few different families: PAC-Bayes (McAllester, 1999; Dziugaite and Roy, 2017; Neyshabur et al., 2017); VC-dimension (Vapnik and Chervonenkis, 1971); and norm-based bounds (Neyshabur et al., 2015b; Bartlett et al., 2017; Neyshabur et al., 2018a). The empirically motivated measures from prior literature that we consider are based on sharpness measure (Keskar et al., 2016); Fisher-Rao measure (Liang et al., 2017); distance of trained weights from initialization (Nagarajan and Kolter, 2019b) and path norm (Neyshabur et al., 2015a). Finally, we consider some optimization based measures based on the speed of the optimization algorithm as motivated by the work of (Hardt et al., 2015) and (Wilson et al., 2017a), and the magnitude of the gradient noise as motivated by the work of (Chaudhari and Soatto, 2018) and (Smith and Le, 2017).

A few papers have explored a large scale study of generalization in deep networks. Neyshabur et al. (2017) perform a small scale study of the generalization of PAC-Bayes, sharpness and a few different norms, and the generalization analysis is restricted to correlation. Jiang et al. (2018) studied the role of margin as a predictor of the generalization gap. However, they used a significantly more restricted set of models (e.g. no depth variations), the experiments were not controlled for potential undesired correlation (e.g. the models can have vastly different training error) and some measures contained parameters that must be learned from the set of models. Novak et al. (2018) conducted large scale study of neural networks but they only looked at correlation of a few measures to generalization. In contrast, we study thousands of models, and perform controlled experiments to avoid undesired artificial correlations. Some of our analysis techniques are inspired by Neal (2019) who proposed the idea of studying generalization in deep models via causal graphs, but did not provide any details or empirical results connected to that idea. Our work focuses on measures that can be computed on a single model and compares a large number of bounds and measures across a much wider range of models in a carefully controlled fashion.

## 2    GENERALIZATION: WHAT IS THE GOAL AND HOW TO EVALUATE?

Generalization is arguably the most fundamental and yet mysterious aspect of machine learning. The core question in generalization is what causes the triplet of a *model*, *optimization* algorithm, and *data properties*[1], to generalize well beyond the training set. There are many hypotheses concerning this question, but what is the right way to compare these hypotheses? The core component of each hypothesis is *complexity measure* that monotonically relates to some aspect of generalization. Here we briefly discuss some potential approaches to compare different complexity measures:

- **Tightness of Generalization Bounds.** Proving generalization bounds is very useful to establish the causal relationship between a complexity measure and the generalization error. However, almost all existing bounds are vacuous on current deep learning tasks (combination of models and datasets), and therefore, one cannot rely on their proof as an evidence on the causal relationship between a complexity measure and generalization currently[2].

- **Regularizing the Complexity Measure.** One may evaluate a complexity measure by adding it as a regularizer and directly optimizing it, but this could fail due to two reasons. The complexity measure could change the loss landscape in non-trivial ways and make the optimization more difficult. In such cases, if the optimization fails to optimize the measure, no conclusion can be made about the causality. Another, and perhaps more critical, problem is the existence of implicit regularization of the optimization algorithm. This makes it hard to run a controlled experiment since one cannot simply turn off the implicit regularization; therefore, if optimizing a measure does not improve generalization it could be simply due to the fact that it is regularizing the model in the same way as the optimization is regularizing it implicitly.

- **Correlation with Generalization** Evaluating measures based on correlation with generalization is very useful but it can also provide a misleading picture. To check the correlation, we should vary architectures and optimization algorithms to produce a set of models. If the set is generated in an artificial way and is not representative of the typical setting, the conclusions might be deceiving and might not generalize to typical cases. One such example is training with different portions of random labels which artificially changes the dataset. Another pitfall is drawing conclusion from changing one or two hyper-parameters (e.g changing the width or batch-size and checking if a measure would correlate with generalization). In these cases, the hyper-parameter could be the true cause of both change in the measure and change in the generalization, but the measure itself has no causal relationship with generalization. Therefore, one needs to be very careful with experimental design to avoid unwanted correlations.

In this work we focus on the third approach. While acknowledging all limitations of a correlation analysis, we try to improve the procedure and capture some of the causal effects as much as possible through careful design of controlled experiments. Further, to evaluate the effectiveness of complexity measures as accurately as possible, we analyze them over sufficiently trained models with a wide range of variations in hyperparameters. For practical reasons, these models must reach convergence within a reasonable time budget. *Details of the notations used are outlined in Appendix A.*

### 2.1    TRAINING MODELS ACROSS HYPERPARAMETER SPACE

In order to create models with different generalization behavior, we consider various hyperparameter types, which are known or believed to influence generalization (e.g. batch size, dropout rate, etc.). Formally, denote each hyperparameter by $\theta_i$ taking values from the set $\Theta_i$, for $i = 1, \ldots, n$ and $n$ denoting the total number of hyperparameter types[3]. For each value of hyperparameters $\boldsymbol{\theta} \triangleq (\theta_1, \theta_2, \ldots, \theta_n) \in \Theta$, where $\Theta \triangleq \Theta_1 \times \Theta_2 \times \cdots \times \Theta_n$, we train the architecture until the training loss (cross-entropy value) reaches a given threshold $\epsilon$. See the Appendix C.2 for a discussion on the choice of the stopping criterion. Doing this for each hyper-parameter configuration $\boldsymbol{\theta} \in \Theta$, we obtain a *total* of $|\Theta|$ models. The space $\Theta$ reflects our prior knowledge about a reasonable hyperparameter

---

[1]For example, it is expected that images share certain structures that allows some models (which leverage these biases) to generalize.

[2]See Dziugaite and Roy (2017) for an example of non-vacuous generalization bound and related discussions.

[3]In our analysis we use $n = 7$ hyperparameters: batch size, dropout probability, learning rate, network depth, weight decay coefficient, network width, optimizer.

space, both in terms of their types and values. Regarding the latter, one could, for example, create $\Theta_i$ by grid sampling of a reasonable number of points within a reasonable range of values for $\theta_i$.

## 2.2 EVALUATION CRITERIA

### 2.2.1 KENDALL'S RANK-CORRELATION COEFFICIENT

One way to evaluate the quality of a complexity measure $\mu$ is through *ranking*. Given a set of models resulted by training with hyperparameters in the set $\Theta$, their associated generalization gap $\{g(\boldsymbol{\theta}) \,|\, \boldsymbol{\theta} \in \Theta\}$, and their respective values of the measure $\{\mu(\boldsymbol{\theta}) \,|\, \boldsymbol{\theta} \in \Theta\}$, our goal is to analyze how consistent a measure (e.g. $\ell_2$ norm of network weights) is with the empirically observed generalization. To this end, we construct a set $\mathcal{T}$, where each element of the set is associated with one of the trained models. Each element has the form of a pair: complexity measure $\mu$ versus generalization gap $g$.

$$\mathcal{T} \triangleq \cup_{\boldsymbol{\theta} \in \Theta} \big\{ \, \big( \mu(\boldsymbol{\theta}), g(\boldsymbol{\theta}) \big) \big\}. \tag{1}$$

An ideal complexity measure must be such that, for any pair of trained models, if $\mu(\boldsymbol{\theta}_1) > \mu(\boldsymbol{\theta}_2)$, then so is $g(\boldsymbol{\theta}_1) > g(\boldsymbol{\theta}_2)$. We use Kendall's rank coefficient $\tau$ (Kendall, 1938) to capture to what degree such consistency holds among the elements of $\mathcal{T}$.

$$\tau(\mathcal{T}) \triangleq \frac{1}{|\mathcal{T}|(|\mathcal{T}| - 1)} \sum_{(\mu_1, g_1) \in \mathcal{T}} \sum_{(\mu_2, g_2) \in \mathcal{T} \setminus (\mu_1, g_1)} \operatorname{sign}(\mu_1 - \mu_2) \operatorname{sign}(g_1 - g_2) \tag{2}$$

Note that $\tau$ can vary between $1$ and $-1$ and attains these extreme values at perfect agreement (two rankings are the same) and perfect disagreement (one ranking is the reverse of the other) respectively. If complexity and generalization are independent, the coefficient becomes zero.

### 2.2.2 GRANULATED KENDALL'S COEFFICIENT

While Kendall's correlation coefficient is an effective tool widely used to capture relationship between 2 rankings of a set of objects, we found that certain measures can achieve high $\tau$ values in a trivial manner – i.e. the measure may strongly correlate with the generalization performance without necessarily capturing the cause of generalization. We will analyze this phenomenon in greater details in subsequent sections. To mitigate the effect of spurious correlations, we propose a new quantity for reflecting the correlation between measures and generalization based on a more controlled setting.

None of the existing complexity measures is perfect. However, they might have different sensitivity and accuracy w.r.t. different hyperparameters. For example, sharpness may do better than other measures when only a certain hyperparameter (say batch size) changes. To understand such details, in addition to $\tau(\mathcal{T})$, we compute $\tau$ for consistency within each hyperparameter axis $\Theta_i$, and then average the coefficient across the remaining hyperparameter space. Formally, we define:

$$m_i \triangleq |\Theta_1 \times \cdots \times \Theta_{i-1} \times \Theta_{i+1} \times \cdots \times \Theta_n| \tag{3}$$

$$\psi_i \triangleq \frac{1}{m_i} \sum_{\theta_1 \in \Theta_1} \cdots \sum_{\theta_{i-1} \in \Theta_{i-1}} \sum_{\theta_{i+1} \in \Theta_{i+1}} \cdots \sum_{\theta_n \in \Theta_n} \tau \big( \cup_{\theta_i \in \Theta_i} \{ \big( \mu(\boldsymbol{\theta}), g(\boldsymbol{\theta}) \big) \} \big) \tag{4}$$

The inner $\tau$ reflects the ranking correlation between the generalization and the complexity measure for a small group of models where the only difference among them is the variation along a single hyperparameter $\theta_i$. We then average the value across all combinations of the other hyperparameter axis. Intuitively, if a measure is good at predicting the effect of hyperparameter $\theta_i$ over the model distribution, then its corresponding $\psi_i$ should be high. Finally, we compute the average $\psi_i$ of average across all hyperparamter axes, and name it $\Psi$:

$$\Psi \triangleq \frac{1}{n} \sum_{i=1}^{n} \psi_i \tag{5}$$

If a measure achieves a high $\Psi$ on a given hyperparameter distribution $\Theta$, then it should achieve high individual $\psi$ across all hyperparameters. A complexity measure that excels at predicting changes in a single hyperparameter (high $\psi_i$) but fails at the other hyperparameters (low $\psi_j$ for all $j \neq i$) will not do well on $\Psi$. On the other hand, if the measure performs well on $\Psi$, it means that the measure can reliably rank the generalization for each of the hyper-parameter changes.

A *thought experiment* to illustrate why $\Psi$ captures a *better* causal nature of the generalization than Kendall's $\tau$ is as follows. Suppose there exists a measure that perfectly captures the depth of the network while producing random prediction if 2 networks have the same depth, this measure would do reasonably well in terms of $\tau$ but much worse in terms of $\Psi$. In the experiments we consider in the following sections, we found that such a measure would achieve overall $\tau = 0.362$ but $\Psi = 0.11$.

### 2.2.3 CONDITIONAL INDEPENDENCE TEST: TOWARDS CAPTURING THE CAUSAL RELATIONSHIPS

Relying on correlation is intuitive but perhaps unsatisfactory. In our experiments, we change several hyper-parameters and assess the correlation between a complexity measure and generalization; however, changing the hyper-parameters can also be seen as *interventions* or *randomized tests* since they are chosen independently from any other variables, and we can use this observation to uncover causal structures. Specifically, we rely on the conditional independent test which measures the conditional mutual information between random variables. Details are outlined in Appendix B.

## 3 GENERATING A FAMILY OF TRAINED MODELS

We chose 7 common hyperparameter types related to optimization and architecture design, with 3 choices for each hyperparameter. We generated $3^7 = 2187$ models that are trained on the CIFAR-10 dataset. We analyze these 2187 models in the subsequent sections; however, additional results including repeating the experiments 5 times as well as training the models using SVHN dataset are presented[4] in Appendix Section C.5. These additional experiments, which add up to more than 10,000 trained models, suggest that the observations we make here are robust to randomness, and, more importantly, captures general behaviors of image classification tasks.

We trained these models to convergence. Convergence criterion is chosen as when cross-entropy loss reaches the value 0.01. Any model that was not able to achieve this value of cross-entropy[5] was discarded from further analysis. The latter is different from the DEMOGEN dataset (Jiang et al., 2018) where the models are not trained to the same cross-entropy. Putting the stopping criterion on *the training loss* rather than *the number of epochs* is crucial since otherwise one can simply use cross-entropy loss value to predict generalization. Please see Appendix Section C.2 for a discussion on the choice of stopping criterion.

To construct a pool of trained models with vastly different generalization behaviors while being able to fit the training set, we covered a wide range of hyperparameters for training. Our base model is inspired by the Network-in-Network (Gao et al., 2011). The hyperparameter categories we test on are: weight decay coefficient (`weight decay`), width of the layer (`width`), mini-batch size (`batch size`), learning rate (`learning rate`), dropout probability (`dropout`), depth of the architecture (`depth`) and the choice of the optimization algorithms (`optimizer`). We select 3 choices for each hyperparameter (i.e. $|\Theta_i| = 3$). Please refer to Appendix C.3 for the details on the models, and Appendix C.1 for the reasoning behind the design choices.

Figure 1 shows summarizing statistics of the models in this study. On the left we show the number of models that achieve above 99% training accuracy for every individual hyperparameter choice. Since we have $3^7 = 2187$ models in total, the maximum number of models for each hyperparameter type is $3^{7-1} = 718$; the majority of the models in our pool were able to reach this threshold. In the middle we show the distribution of the cross-entropy value over the entire training set. While we want the models to be at exactly 0.01 cross-entropy, in practice it is computationally prohibitive to constantly evaluate the loss over the entire training set; further, to enable reasonable temporal granularity, we estimate the training loss with 100 randomly sampled minibatch. These computational compromises result in long-tailed distribution of training loss centered at 0.01. As shown in Table 1, even such minuscule range of cross-entropy difference could lead to positive correlation with generalization, highlighting the importance of training loss as a stopping criterion. On the right, we show the distribution of the generalization gap. Notice while all the models' training accuracy is above 0.99, there is a wide range of generalization gap, which is ideal for evaluating complexity measures.

---

[4]All the experiments reported in the main text have been repeated for 5 times. The mean (Table 9) is consistent with those presented in the main text and standard deviation (Table 10) is very small compared to the magnitude of the mean for all measures. Further, we also repeat the experiments once on the SVHN dataset (Table 7), whose results are also consistent with the observations made on CIFAR-10.

[5]In our analysis, less than 5 percent of the models do not reach this threshold.

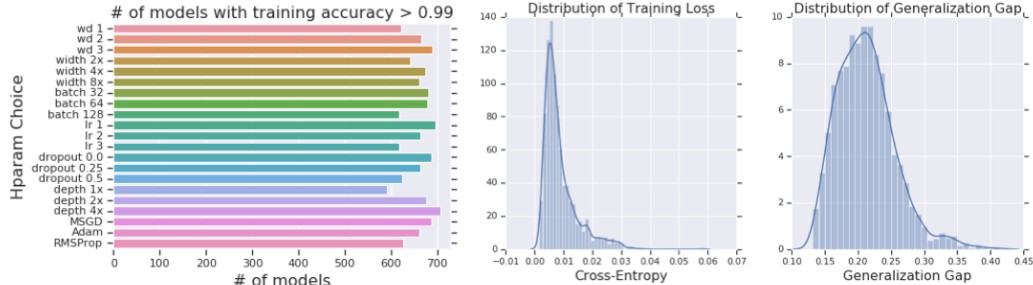

Figure 1: **Left**: Number of models with training accuracy$> 99\%$ for each hyperparameter type. **Mid.**: Distribution of training cross-entropy (that of training error in Fig. 4). **Right**: Distribution of generalization gap.

## 4 PERFORMANCE OF COMPLEXITY MEASURES

### 4.1 BASELINE COMPLEXITY MEASURES

The first baseline we consider is performance of a measure against an oracle who observes the noisy generalization gap. Concretely, we rank the models based on the true generalization gap with some additive noise. The resulting ranking correlation indicates how close the performances of all models are. As the scale of the noise approaches 0, the oracle's prediction tends towards perfect (i.e. 1). This baseline accounts for the potential noise in the training procedure and gives an anchor for gauging the difficulty of each hyperparameter type. Formally, given an arbitrary set of hyper-parameters $\Theta'$, we define $\epsilon$-oracle to be the expectation of $\tau$ or $\Psi$ where the measure is $\{g(\boldsymbol{\theta}) + \mathcal{N}(0, \epsilon^2) \,|\, \boldsymbol{\theta} \in \Theta'\}$. We report the performance of the noisy oracle in Table 1 for $\epsilon \in \{0.02, 0.05\}$. For additional choices of $\epsilon$ please refer to Appendix C.5.

Second, to understand how our hyperparameter choices affect the optimization, we give each hyperparameter type a *canonical order* which is believed to have correlation with generalization (e.g. larger learning rate generalizes better) and measure their $\tau$. The exact canonical ordering can be found in Appendix C.4. Note that unlike other measures, each canonical ordering can only predict generalization for its own hyperparameter type, since its corresponding hyperparameter remains fixed in any other hyperparameter type; consequently, each column actually represents different measure for the *canonical measure* row. Assuming that each canonical measure is uninformative of any other canonical measures, the $\Psi$ criterion for each canonical measure is $\frac{1}{7}$ of its performance on the corresponding hyperparameter type.

|      |                    | batch size | dropout | learning rate | depth  | optimizer | weight decay | width  | overall $\tau$ | $\Psi$ |
|------|--------------------|------------|---------|---------------|--------|-----------|--------------|--------|----------------|--------|
| Corr | vc dim 19          | 0.000      | 0.000   | 0.000         | -0.909 | 0.000     | 0.000        | -0.171 | -0.251         | -0.154 |
|      | # params 20        | 0.000      | 0.000   | 0.000         | -0.909 | 0.000     | 0.000        | -0.171 | -0.175         | -0.154 |
|      | $1/\gamma$ (22)    | 0.312      | -0.593  | 0.234         | 0.758  | 0.223     | -0.211       | 0.125  | 0.124          | 0.121  |
|      | entropy 23         | 0.346      | -0.529  | 0.251         | 0.632  | 0.220     | -0.157       | 0.104  | 0.148          | 0.124  |
|      | cross-entropy 21   | 0.440      | -0.402  | 0.140         | 0.390  | 0.149     | 0.232        | 0.080  | 0.147          | 0.147  |
|      | oracle 0.02        | 0.380      | 0.657   | 0.536         | 0.717  | 0.374     | 0.388        | 0.360  | 0.714          | 0.487  |
|      | oracle 0.05        | 0.172      | 0.375   | 0.305         | 0.384  | 0.165     | 0.184        | 0.204  | 0.438          | 0.256  |
|      | canonical ordering | 0.652      | 0.969   | 0.733         | 0.909  | -0.055    | 0.735        | 0.171  | N/A            | N/A    |
|      |                    |            |         |               |        |           |              |        | $|\mathcal{S}| = 2$ | $\min \forall |\mathcal{S}|$ |
| MI   | vc dim             | 0.0422     | 0.0564  | 0.0518        | 0.0039 | 0.0422    | 0.0443       | 0.0627 | 0.00           | 0.00   |
|      | # param            | 0.0202     | 0.0278  | 0.0259        | 0.0044 | 0.0208    | 0.0216       | 0.0379 | 0.00           | 0.00   |
|      | $1/\gamma$         | 0.0108     | 0.0078  | 0.0133        | 0.0750 | 0.0105    | 0.0119       | 0.0183 | 0.0051         | 0.0051 |
|      | entropy            | 0.0120     | 0.0656  | 0.0113        | 0.0086 | 0.0120    | 0.0155       | 0.0125 | 0.0065         | 0.0065 |
|      | cross-entropy      | 0.0233     | 0.0850  | 0.0118        | 0.0075 | 0.0159    | 0.0119       | 0.0183 | 0.0040         | 0.0040 |
|      | oracle 0.02        | 0.4077     | 0.3557  | 0.3929        | 0.3612 | 0.4124    | 0.4057       | 0.4154 | 0.1637         | 0.1637 |
|      | oracle 0.05        | 0.1475     | 0.1167  | 0.1369        | 0.1241 | 0.1515    | 0.1469       | 0.1535 | 0.0503         | 0.0503 |
|      | random             | 0.0005     | 0.0002  | 0.0005        | 0.0002 | 0.0003    | 0.0006       | 0.0009 | 0.0004         | 0.0001 |

Table 1: Numerical Results for Baselines and Oracular Complexity Measures

We next look at one of the most well-known complexity measures in machine learning; the VC-Dimension. Bartlett et al. (2019) proves bounds on the VC dimension of piece-wise linear networks with potential weight sharing. In Appendix D.1, we extend their result to include pooling layers and multi-class classification. We report two complexity measures based on VC-dimension bounds and parameter counting. These measures could be predictive merely when the architecture changes, which happens only in `depth` and `width` hyperparameter types. We observe that, with both types,

VC-dimension as well as the number of parameters are ***negatively correlated*** with generalization gap which *confirms the widely known empirical observation that overparametrization improves generalization in deep learning*.

Finally, we report the measures that only look at the output of the network. In particular, we look at the cross-entropy loss, margin $\gamma$, and the entropy of the output. These three measures are closely related to each other. In fact, the outcomes in Table 1 reflects this similarity. These results *confirm the general understanding that larger margin, lower cross-entropy and higher entropy would lead to better generalization*. Please see Appendix D.1.1 for definitions and more discussions on these measures.

## 4.2 SURPRISING FAILURE OF SOME (NORM & MARGIN)-BASED MEASURES

In machine learning, a long standing measure for quantifying the complexity of a function, and therefore generalization, is using some norm of the given function. Indeed, directly optimizing some of the norms can lead to improved generalization. For example, $\ell_2$ regularization on the parameters of a model can be seen as imposing an isotropic Gaussian prior over the parameters in maximum a posteriori estimation. We choose several representative norms (or measures based on norms) and compute our correlation coefficient between the measures and the generalization gap of the model.

We study the following measures and their variants (Table 2): *spectral bound*, *Frobenius distance from initialization*, $\ell_2$ *Frobenius norm of the parameters*, *Fisher-Rao metric* and *path norm*.

| | | batch size | dropout | learning rate | depth | optimizer | weight decay | width | overall $\tau$ | $\Psi$ |
|---|---|---|---|---|---|---|---|---|---|---|
| **Corr** | Frob distance 40 | -0.317 | -0.833 | -0.718 | 0.526 | -0.214 | -0.669 | -0.166 | -0.263 | -0.341 |
| | Spectral orig 26 | -0.262 | -0.762 | -0.665 | -0.908 | -0.131 | -0.073 | -0.240 | -0.537 | -0.434 |
| | Parameter norm 42 | 0.236 | -0.516 | 0.174 | 0.330 | **0.187** | 0.124 | -0.170 | 0.073 | 0.052 |
| | Path norm 44 | 0.252 | **0.270** | 0.049 | **0.934** | 0.153 | 0.338 | **0.178** | **0.373** | **0.311** |
| | Fisher-Rao 45 | **0.396** | 0.147 | **0.240** | -0.553 | 0.120 | **0.551** | 0.177 | 0.078 | 0.154 |
| | oracle 0.02 | 0.380 | 0.657 | 0.536 | 0.717 | 0.374 | 0.388 | 0.360 | 0.714 | 0.487 |
| | | | | | | | | | $\|\mathcal{S}\| = 2$ | $\min \forall \|\mathcal{S}\|$ |
| **MI** | Frob distance | 0.0462 | 0.0530 | 0.0196 | **0.1559** | 0.0502 | 0.0379 | 0.0506 | 0.0128 | 0.0128 |
| | Spectral orig | **0.2197** | **0.2815** | **0.2045** | 0.0808 | **0.2180** | **0.2285** | **0.2181** | **0.0359** | **0.0359** |
| | Parameter norm | 0.0039 | 0.0197 | 0.0066 | 0.0115 | 0.0064 | 0.0049 | 0.0167 | 0.0047 | 0.0038 |
| | Path norm | 0.1027 | 0.1230 | 0.1308 | 0.0315 | 0.1056 | 0.1028 | 0.1160 | 0.0240 | 0.0240 |
| | Fisher Rao | 0.0060 | 0.0072 | 0.0020 | 0.0713 | 0.0057 | 0.0014 | 0.0071 | 0.0018 | 0.0013 |
| | oracle 0.05 | 0.1475 | 0.1167 | 0.1369 | 0.1241 | 0.1515 | 0.1469 | 0.1535 | 0.0503 | 0.0503 |

Table 2: Numerical Results for Selected (Norm & Margin)-Based Complexity Measures

**Spectral bound**: The most surprising observation here is that the *spectral complexity is strongly negatively correlated with generalization*, and negatively correlated with changes within every hyperparameter type. Most notably, it has strong negative correlation with the *depth* of the network, which may suggest that the largest singular values are not sufficient to capture the capacity of the model. To better understand the reason behind this observation, we investigate using different components of the spectral complexity as the measure. An interesting observation is that the Frobenius distance to initialization is negatively correlated, but the Frobenius norm of the parameters is slightly positively correlated with generalization, which contradicts some theories suggesting solutions closer to initialization should generalize better. A tempting hypothesis is that weight decay favors solution closer to the origin, but we did an ablation study on only models with 0 weight decay and found that the distance from initialization still correlates negatively with generalization.

These observations correspond to choosing different reference matrices $\mathbf{W}_i^0$ for the bound: the distance corresponds to using the initialization as the reference matrices while the Frobenius norm of the parameters corresponds to using the origin as the reference. Since the Frobenius norm of the parameters shows better correlation, we use zero reference matrices in the spectral bound. This improved both $\tau$ and $\Psi$, albeit still negative. In addition, we extensively investigate the effect of different terms of the Spectral bound to isolate the effect; however, the results do not improve. These experiments can be found in the Appendix D.2.

**Path norm**: While path-norm is a proper norm in the function space but not in parameter space, we observe that it is positively correlated with generalization in all hyper-parameter types and achieves comparable $\tau$ (0.373) and $\Psi$ (0.311).

**Fisher-Rao metric**: The Fisher-Rao metric is a lower bound (Liang et al., 2017) on the path norm that has been recently shown to capture generalization. We observed that it overall shows worse

correlation than the path norm; in particular, it is negatively correlated with the depth of the network, which contrasts with path norm that properly captures the effect of depth on generalization. A more interesting observation is that the Fisher-Rao metric achieves a positive $\Psi = 0.154$ but its $\tau = 0.078$ is essentially at chance. This may suggest that the metric *can capture a single hyper-parameter change but is not able to capture the interactions between different hyperparameter types*.

**Effect of Randomness**: `dropout` and `batch size` (first 2 columns of Table 2) directly introduce randomness into the training dynamic. For batch size, we observed that the Frobenius displacement and spectral complexity both correlate negatively with the changes in batch size while the Frobenius norm of the parameters correlates positively with generalization. On the other hand, when changes happen to the magnitude dropout probability, we observed that all of the proper norms are negatively correlated with the generalization changes. Since increasing dropout usually reduces the generalization gap, this implies that increasing the dropout probability may be at least partially responsible for the growth in these norms. This is unexpected since increasing norm in principle implies higher model capacity which is usually more prone to overfitting.

The overall picture does not change much going from the ranking correlation to mutual information, with a notable exception where spectral complexity has the highest conditional mutual information compared to all the other measures. This is due to the fact that the conditional mutual information is agnostic to the direction of correlation, and in the ranking correlation, spectral complexity has the highest absolute correlation. While this view might seem contradictory to classical view as the spectral complexity is a complexity measure which should be small to guarantee good generalization, it is nonetheless informative about the generalization of the model. Further, by inspecting the conditional mutual information for each hyperparameter, we find that the majority of spectral complexity's predictive power is due to its ability to capture the depth of the network, as the mutual information is significantly lower if depth is already observed.

### 4.3 SUCCESS OF SHARPNESS-BASED MEASURES

A natural category of generalization measures is centered around the concept of "*sharpness*" of the local minima, capturing the sensitivity of the empirical risk (i.e. the loss over the entire training set) to perturbations in model parameters. Such notion of stability under perturbation is captured elegantly by the PAC-Bayesian framework (McAllester, 1999) which has provided promising insights for studying generalization of deep neural networks (Dziugaite and Roy, 2017; Neyshabur et al., 2017; 2018a). In this sections, we investigate PAC-Bayesian generalization bounds and several of their variants which rely on different priors and different notions of sharpness (Table 3).

In order to evaluate a PAC-Bayesian bound, one needs to come up with a prior distribution over the parameters that is chosen in advance before observing the training set. Then, given any posterior distribution on the parameters which could depend on the training set, a PAC-Bayesian bound (Theorem 46) states that the expected generalization error of the parameters generated from the posterior can be bounded by the KL-divergence of the prior and posterior. The posterior distribution can be seen as adding perturbation on final parameters. Dziugaite and Roy (2017) shows contrary to other generalization bounds, it is possible to calculate non-vacuous PAC-Bayesian bounds by optimizing the bound over a large set of Gaussian posteriors. Neyshabur et al. (2017) demonstrates that when prior and posterior are isotropic Gaussian distributions, then PAC-Bayesian bounds are good measure of generalization on small scale experiments; see Eq (47).

PAC-Bayesian framework captures sharpness in the *expected* sense since we add randomly generated perturbations to the parameters. Another possible notion of sharpness is the *worst-case* sharpness where we search for the direction that changes the loss the most. This is motivated by (Keskar et al., 2016) where they observe that this notion would correlate with generalization in the case of different batch sizes. We can use PAC-Bayesian framework to construct generalization bounds for this worst-case perturbations as well. We refer to this worst case bound as the sharpness bound in Eq (50). The main component in both PAC-Bayes and worst-case sharpness bounds is the ratio of norm of parameters to the magnitude of the perturbation, where the magnitude is chosen to be the largest number such that the training error of the perturbed model is at most $0.1$. While mathematically, the sharpness bound should always yield higher complexity than the PAC-Bayes bound, we observed that the former has higher correlation both in terms of $\tau$ and $\Psi$. In addition, we studied inverse of perturbation magnitude as a measure by removing the norm in the numerator to compare it with the bound. However, we did not observe a significant difference.

| | | batch size | dropout | learning rate | depth | optimizer | weight decay | width | overall $\tau$ | $\Psi$ |
|---|---|---|---|---|---|---|---|---|---|---|
| **Corr** | sharpness-orig 52 | 0.542 | -0.359 | 0.716 | 0.816 | 0.297 | 0.591 | 0.185 | 0.400 | 0.398 |
| | pacbayes-orig 49 | 0.526 | -0.076 | 0.705 | 0.546 | **0.341** | 0.564 | -0.086 | 0.293 | 0.360 |
| | $1/\alpha'$ sharpness mag 62 | **0.570** | **0.148** | **0.762** | 0.824 | 0.297 | **0.741** | **0.269** | **0.484** | **0.516** |
| | $1/\sigma'$ pacbayes mag 61 | 0.490 | -0.215 | 0.505 | **0.896** | 0.186 | 0.147 | 0.195 | 0.365 | 0.315 |
| | oracle 0.02 | 0.380 | 0.657 | 0.536 | 0.717 | 0.374 | 0.388 | 0.360 | 0.714 | 0.487 |
| | | | | | | | | | $|\mathcal{S}| = 2$ | $\min \forall |\mathcal{S}|$ |
| **MI** | sharpness-orig | 0.1117 | 0.2353 | 0.0809 | 0.0658 | 0.1223 | 0.1071 | 0.1254 | 0.0224 | 0.0224 |
| | pacbayes-orig | 0.0620 | 0.1071 | 0.0392 | 0.0597 | 0.0645 | 0.0550 | 0.0977 | 0.0225 | 0.0225 |
| | $1/\alpha'$ sharpness mag | **0.1640** | **0.2572** | **0.1228** | **0.1424** | **0.1779** | **0.1562** | **0.1786** | **0.0544** | **0.0544** |
| | $1/\sigma'$ pacbayes mag | 0.0884 | 0.1514 | 0.0813 | 0.0399 | 0.1004 | 0.1025 | 0.0986 | 0.0241 | 0.0241 |
| | oracle 0.05 | 0.1475 | 0.1167 | 0.1369 | 0.1241 | 0.1515 | 0.1469 | 0.1535 | 0.0503 | 0.0503 |

Table 3: Numerical results for selected Sharpness-Based Measures; all the measure use the origin as the reference and `mag` refers to magnitude-aware version of the measure.

### 4.3.1 MAGNITUDE-AWARE PERTURBATION BOUNDS

Perturbing the parameters without taking their magnitude into account can cause many of them to switch signs. Therefore, one cannot apply large perturbations to the model without changing the loss significantly. One possible modification to improve the perturbations is to choose the perturbation magnitude based on the magnitude of the parameter. In that case, it is guaranteed that if the magnitude of perturbation is less than the magnitude of the parameter, then the sign of the parameter does not change. Following Keskar et al. (2016), we pick the magnitude of the perturbation with respect to the magnitude of parameters. We formalize this notion of importance based magnitude. Specifically, we derive two alternative generalization bounds for expected sharpness in Eq (55) and worst case sharpness in Eq (58) that include the magnitude of the parameters into the prior. Formally, we design $\alpha'$ and $\sigma'$, respectively for sharpness and PAC-Bayes bounds, to be the ratio of parameter magnitude to the perturbation magnitude. While this change did not improve upon the original PAC-Bayesian measures, we observed that simply looking at $1/\alpha'$ has *surprising predictive power* in terms of the generalization which surpasses the performance of oracle 0.02. This measure is very close to what was originally suggested in Keskar et al. (2016). Its effectiveness is further corroborated by the conditional mutual information based metric, where we observed that $1/\alpha'$ has the highest mutual information with generalization among all hyperparameters and also overall.

### 4.3.2 FINDING $\sigma$

In case of models with extremely small loss, the perturbed loss should roughly increase monotonically with respect to the perturbation scale. Leveraging this observation, we design algorithms for computing the perturbation scale $\sigma$ such that the first term on the RHS is as close to a fixed value as possible for all models. In our experiments, we choose the deviation to be 0.1 which translates to 10% training error. These search algorithms are paramount to compare measures between different models. We provide the detailed algorithms in the Appendix E. To improve upon our algorithms, one could try a computational approach similar to Dziugaite and Roy (2017) to obtain a numerically better bound which may result in stronger correlation. However, due to practical computational constraints, we could not do so for the large number of models we consider.

### 4.4 POTENTIAL OF OPTIMIZATION-BASED MEASURES

Optimization is an indispensable component of deep learning. Numerous optimizers have been proposed for more stable training and faster convergence. How the optimization scheme and speed of optimization influence generalization of a model has been a topic of contention among the deep learning community (Merity et al., 2017; Hardt et al., 2015). We study 3 representative optimizers `Momentum SGD`, `Adam`, and `RMSProp` with different initial learning rates in our experiments to thoroughly evaluate this phenomenon. We also consider other optimization related measures that are believed to correlate with generalization. These include (Table 4):

1. Number of iterations required to reach cross-entropy equals 0.1

2. Number of iterations required going from cross-entropy equals 0.1 to cross-entropy equals 0.01

3. Variance of the gradients after only seeing the entire dataset once (1 epoch)

4. Variance of the gradients when the cross-entropy is approximately 0.01

| | | batch size | dropout | learning rate | depth | optimizer | weight decay | width | overall $\tau$ | $\Psi$ |
|---|---|---|---|---|---|---|---|---|---|---|
| **Corr** | step to 0.1 63 | -0.664 | -0.861 | -0.255 | **0.440** | -0.030 | -0.628 | 0.043 | -0.264 | -0.279 |
| | step 0.1 to 0.01 64 | -0.151 | -0.069 | -0.014 | 0.114 | 0.072 | -0.046 | -0.021 | -0.088 | -0.016 |
| | grad noise 1 epoch 65 | 0.071 | **0.378** | 0.376 | -0.517 | 0.121 | 0.221 | 0.037 | 0.070 | 0.098 |
| | grad noise final 66 | **0.452** | 0.119 | **0.427** | 0.141 | **0.245** | **0.432** | **0.230** | **0.311** | **0.292** |
| | oracle 0.02 | 0.380 | 0.657 | 0.536 | 0.717 | 0.374 | 0.388 | 0.360 | 0.714 | 0.487 |
| | | | | | | | | | $|\mathcal{S}| = 2$ | $\min \forall |\mathcal{S}|$ |
| **MI** | step to 0.1 | 0.0349 | 0.0361 | 0.0397 | **0.1046** | 0.0485 | 0.0380 | 0.0568 | 0.0134 | 0.0134 |
| | step 0.1 to 0.01 | 0.0125 | 0.0031 | 0.0055 | 0.0093 | 0.0074 | 0.0043 | 0.0070 | 0.0032 | 0.0032 |
| | grad noise 1 epoch | 0.0051 | 0.0016 | 0.0028 | 0.0633 | 0.0113 | 0.0027 | 0.0052 | 0.0013 | 0.0013 |
| | grad noise final | **0.0623** | **0.0969** | **0.0473** | 0.0934 | **0.0745** | **0.0577** | **0.0763** | **0.0329** | **0.0329** |
| | oracle 0.05 | 0.1475 | 0.1167 | 0.1369 | 0.1241 | 0.1515 | 0.1469 | 0.1535 | 0.0503 | 0.0503 |

Table 4: Optimization-Based Measures

**Number of Iterations:** The number of iterations roughly characterizes the speed of optimization, which has been argued to correlate with generalization. For the models considered here, we observed that *the initial phase (to reach cross-entropy value of 0.1) of the optimization is negatively correlated with the speed of optimization for both $\tau$ and $\Psi$*. This would suggest that the difficulty of optimization during the initial phase of the optimization benefits the final generalization. On the other hand, the speed of optimization going from cross-entropy 0.1 to cross-entropy 0.01 does not seem to be correlated with the generalization of the final solution. Importantly, the speed of optimization is not an explicit capacity measure so *either positive or negative correlation could potentially be informative*.

**Variance of Gradients:** *Towards the end of the training, the variance of the gradients also captures a particular type of "flatness" of the local minima. This measure is surprisingly predictive of the generalization both in terms of $\tau$ and $\Psi$, and more importantly, is positively correlated across every type of hyperparameter*. To the best of our knowledge, this is the first time this phenomenon has been observed. The connection between variance of the gradient and generalization is perhaps natural since much of the recent advancement in deep learning such as residual networks (He et al., 2016) or batch normalization have enabled using larger learning rates to train neural networks. Stability with higher learning rates implies smaller noises in the minibatch gradient. With the mutual information metric, the overall observation is consistent with that of ranking correlation, but the final gradient noise also outperforms gradient noise at 1 epoch of training conditioned on the dropout probability. We hope that our work encourages future works in other possible measures based on optimization and during training.

## 5 CONCLUSION

We conducted large scale experiments to test the correlation of different measures with the generalization of deep models and propose a framework to better disentangle the cause of correlation from spurious correlation. We confirmed the effectiveness of the PAC-Bayesian bounds through our experiments and corroborate it as a promising direction for cracking the generalization puzzle. Further, we provide an extension to existing PAC-Bayesian bounds that consider the importance of each parameter. We also found that several measures related to optimization are surprisingly predictive of generalization and worthy of further investigation. On the other hand, several surprising failures about the norm-based measures were uncovered. In particular, we found that regularization that introduces randomness into the optimization can increase various norm of the models and spectral complexity related norm-based measures are unable to capture generalization – in fact, most of them are negatively correlated. Our experiments demonstrate that the study of generalization measure can be misleading when the number of models studied is small and the metric of quantifying the relationship is not carefully chosen. We hope this work will incentivize more rigorous treatment of generalization measures in future work.

To the best of our knowledge, this work is one of the most comprehensive study of generalization to date, but there are a few short-comings. Due to computational constraints, we were only able to study 7 most common hyperparameter types and relatively small architectures, which do not reflect the models used in production. Indeed, if more hyperparameters are considered, one could expect to better capture the causal relationship. We also only studied models trained on two image datasets (CIFAR-10 and SVHN), only classification models and only convolutional networks. We hope that future work would address these limitations.

## ACKNOWLEDGEMENT

We thank our colleagues at Google: Guy Gur-Ari for many insightful discussions that helped with the experiment design, Ethan Dyer, Pierre Foret, Sergey Ioffe for their feedback, and Scott Yak for help with implementation. We are grateful for insightful discussions with Brady Neal of University of Montreal about limitation of correlation analysis. We also thank Daniel Roy of University of Toronto for insightful comments.

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

## A    NOTATION

We denote a probability distribution as $\mathscr{A}$, set as $\mathcal{A}$, tensor as $\mathbf{A}$, vector as $\mathbf{a}$, and scalar as $a$ or $\alpha$. Let $\mathscr{D}$ denote the data distributions over inputs and their labels, and let $\kappa$ denote number of classes. We use $\triangleq$ for equality by definition. We denote by $\mathcal{S}$ a given dataset, consisting of $m$ i.i.d tuples $\{(\mathbf{X}_1, y_1), \ldots, (\mathbf{X}_m, y_m)\}$ drawn from $\mathscr{D}$ where $\mathbf{X}_i \in \mathcal{X}$ is the input data and $y_i \in \{1, \ldots, \kappa\}$ the corresponding class label. We denote a feedforward neural network by $f_{\mathbf{w}} : \mathcal{X} \to \mathbb{R}^\kappa$, its weight parameters by $\mathbf{w}$, and the number of weights by $\omega \triangleq \dim(\mathbf{w})$. No activation function is applied at the output (i.e. logits). Denote the weight tensor of the $i^{th}$ layer of the network by $\mathbf{W}_i$, so that $\mathbf{w} = \mathrm{vec}(\mathbf{W}_1, \ldots, \mathbf{W}_d)$, where $d$ is the depth of the network, and $\mathrm{vec}$ represents the vectorization operator. Furthermore, denote by $f_{\mathbf{w}}(\mathbf{X})[j]$ the $j$-th output of the function $f_{\mathbf{w}}(\mathbf{X})$.

Let $\mathcal{R}$ be the set of binary relations, and $I : \mathcal{R} \to \{0, 1\}$ be the indicator function that is 1 if its input is true and zero otherwise. Let $L$ be the 1-0 classification loss over the data distribution $\mathscr{D}$: $L(f_{\mathbf{w}}) \triangleq \mathbb{E}_{(\mathbf{X}, y) \sim \mathscr{D}} \left[ I\big(f_{\mathbf{w}}(\mathbf{X})[y] \leq \max_{j \neq y} f_{\mathbf{w}}(\mathbf{X})[j]\big) \right]$ and let $\hat{L}$ be the empirical estimate of 1-0 loss over $\mathcal{S}$: $\hat{L}(f_{\mathbf{w}}) \triangleq \frac{1}{m} \sum_{i=1}^m I\big(f_{\mathbf{w}}(\mathbf{X})[y_i] \leq \max_{j \neq y_i} f_{\mathbf{w}}(\mathbf{X})[j]\big)$. We refer to $L(f_{\mathbf{w}}) - \hat{L}(f_{\mathbf{w}})$ as the ***generalization gap***. For any input $\mathbf{X}$, we define the sample dependent margin[6] as $\gamma(\mathbf{X}) \triangleq \big(f_{\mathbf{w}}(\mathbf{X})\big)[y] - \max_{i \neq y} f_{\mathbf{w}}(\mathbf{X})_i$. Moreover, we define the overall margin $\gamma$ as the $10^{th}$ percentile (a robust surrogate for the minimum) of $\gamma(\mathbf{X})$ over the entire training set $\mathcal{S}$.

Given any margin value $\gamma \geq 0$, we define the margin loss $L_\gamma$ as follows:

$$L_\gamma(f_{\mathbf{w}}) \triangleq \mathbb{E}_{(\mathbf{X}, y) \sim \mathscr{D}} \left[ I\big(f_{\mathbf{w}}(\mathbf{X})[y] \leq \gamma + \max_{j \neq y} f_{\mathbf{w}}(\mathbf{X})[j]\big) \right] \tag{6}$$

and $\hat{L}_\gamma$ is defined in an analogous manner on the training set. Further, for any vector $\mathbf{v}$, we denote by $\|\mathbf{v}\|_2$ the $\ell_2$ norm of $\mathbf{v}$. For any tensor $\mathbf{W}$, let $\|\mathbf{W}\|_F \triangleq \|\mathrm{vec}(\mathbf{W})\|$. We also denote $\|\mathbf{W}\|_2$ as the spectral norm of the tensor $\mathbf{W}$ when used with a convolution operator. For convolutional operators, we compute the true singular value with the method proposed by Sedghi et al. (2018) through FFT.

We denote a tensor as $\mathbf{A}$, vector as $\boldsymbol{a}$, and scalar as $A$ or $a$. For any $1 \leq j \leq k$, consider a $k$-th order tensor $\mathbf{A}$ and a $j$-th order tensor $\mathbf{B}$ where dimensions of $\mathbf{B}$ match the last $j$ dimensions of $\mathbf{A}$. We then define the product operator $\otimes_j$:

$$(\mathbf{A} \otimes_j \mathbf{B})_{i_1, \ldots, i_{k-j}} \triangleq \langle \mathbf{A}_{i_1, \ldots, i_{k-j}}, \mathbf{B} \rangle, \tag{7}$$

where $i_1, \ldots, i_{k-j}$ are indices. We also assume that the input images have dimension $n \times n$ and there are $\kappa$ classes. Given the number of input channels $c_{\mathrm{in}}$, number of output channels $c_{\mathrm{out}}$, 2D square kernel with side length $k$, stride $s$, and padding $p$, we define the convolutional layer $\mathrm{conv}_{\mathbf{W}, s, p}$ as follows:

$$\mathrm{conv}_{\mathbf{W}, s, p}(\mathbf{X})_{i_1, i_2} \triangleq \mathbf{W} \otimes_3 \mathrm{patch}_{s(i_1 - 1) + 1, s(i_2 - 1) + 1, k}\big(\mathrm{pad}_p(\mathbf{X})\big) \qquad \forall 1 \leq i_1, i_2 \leq \lfloor \frac{n + 2p - k}{s} \rfloor \tag{8}$$

where $\mathbf{W} \in \mathbb{R}^{c_{\mathrm{out}} \times c_{\mathrm{in}} \times k \times k}$ is the convolutional parameter tensor, $\mathrm{patch}_{i,j,k}(\mathbf{Z})$ is a $k \times k$ patch of $\mathbf{Z}$ starting from the point $(i, j)$, and $\mathrm{pad}_p$ is the padding operator which adds $p$ zeros to top, bottom, left and right of $\mathbf{X}$:

$$\mathrm{pad}_p(\mathbf{X})_{i_1, i_2, j} = \begin{cases} \mathbf{X}_{i_1, i_2} & p < i_1, i_2 \leq n + p \\ 0 & \text{otherwise} \end{cases}. \tag{9}$$

We also define the max-pooling operator $\mathrm{pool}_{k, s, p}$ as follows:

$$\mathrm{pool}_{k, s, p}(\mathbf{X})_{i_1, i_2, j} = \max(\mathrm{patch}_{s(i_1 - 1) + 1, s(i_2 - 1) + 1}\big(\mathrm{pad}_p(\mathbf{X}_{:, :, j})\big)) \qquad \forall 1 \leq i_1, i_2 \leq \lfloor \frac{n + 2p - k}{s} \rfloor \tag{10}$$

---

[6]This work only concerns with the output margins, but generally margin can be defined at any layer of a deep network as introduced in (Elsayed et al., 2018) and used to establish a generalization bound in, (Wei and Ma, 2019b).

We denote by $f_{\mathbf{W},s}$ a convolutional network such that $\mathbf{W}_i \in \mathbb{R}^{c_i \times c_{i-1} \times k_i \times k_i}$ is the convolution tensor and $s_i$ is the convolutional stride at layer $i$. At Layer $i$, we assume the sequence of convolution, ReLU and max-pooling where the max pooling has kernel $k_i'$ and stride $s_i'$. Lack of max-pooling in some layers can be achieved by setting $k_i' = s_i' = 1$. We consider classification tasks and denote the number of classes by $\kappa$.

## B  CONDITIONAL INDEPENDENCE TEST: TOWARDS CAPTURING THE CAUSAL RELATIONSHIPS

Relying on correlation is intuitive but perhaps unsatisfactory. In our experiments, we change several hyper-parameters and assess the correlation between a complexity measure and generalization. When we observe correlation between a complexity measure and generalization, we want to differentiate the following two scenarios:

- Changing a hyper-parameter causes the complexity measure to be low and lower value of the measure causes the generalization gap to be low.
- Changing a hyper-parameter causes the complexity measure to be low and changing the same hyper-parameter also causes the generalization to be low but the lower value of the complexity measure by itself has no effect on generalization.

The above two scenarios are demonstrated in Figure 2-Middle and Figure 2-Right respectively. In attempt to truly understand these relationships, we will rely on the tools from probabilistic causality. Our approach is inspired by the seminal work on Inductive Causation (IC) Algorithm by Verma and Pearl (1991), which provides a framework for learning a graphical model through conditional independence test. While the IC algorithm traditionally initiates the graph to be fully connected, we will take advantage of our knowledge about generalization and prune edges of the initialized graph to expedite the computations. Namely, we assume that the choice of hyperparameter does not directly explain generalization, but rather it induces changes in some measure $\mu$ which can be used to explain generalization.

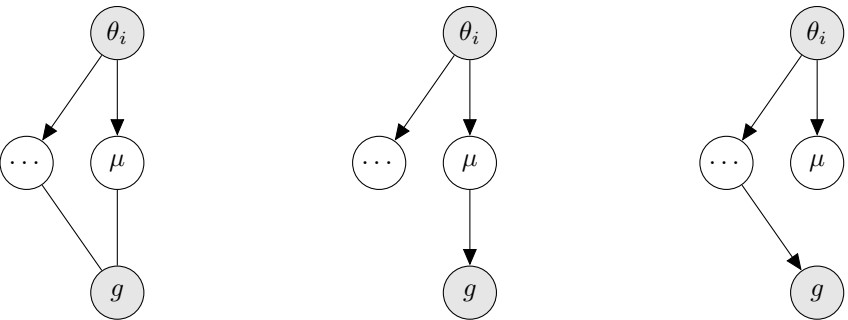

Figure 2: **Left**: Graph at initialization of IC algorithm. **Middle**: The ideal graph where the measure $\mu$ can directly explain observed generalization. **Right**: Graph for correlation where $\mu$ cannot explain observed generalization.

Our primary interest is to establish the existence of an edge between $\mu$ and $g$. Suppose there exists a large family of complexity measures and among them there is a true complexity measure that can fully explain generalization. Then to verify the existence of the edge between $\mu$ and $g$, we can perform the ***conditional independent test*** by reading the conditional mutual information between $\mu$ and $g$ given that a set of hyperparameter types $\mathcal{S}$ is observed[7]. For any function $\phi : \Theta \to \mathbb{R}$, let $V_\phi : \Theta_1 \times \Theta_2 \to \{+1, -1\}$ be as $V_\phi(\theta_1, \theta_2) \triangleq \text{sign}(\phi(\theta_1) - \phi(\theta_2))$. Furthermore, let $U_\mathcal{S}$ be a random variable that correspond to the values of hyperparameters in $\mathcal{S}$. We calculate the conditional mutual information as follows:

$$\mathcal{I}(V_\mu, V_g \,|\, U_\mathcal{S}) = \sum_{U_\mathcal{S}} p(U_\mathcal{S}) \sum_{V_\mu \in \{\pm 1\}} \sum_{V_g \in \{\pm 1\}} p(V_\mu, V_g \,|\, U_\mathcal{S}) \log \left( \frac{p(V_\mu, V_g \,|\, U_\mathcal{S})}{p(V_\mu \,|\, U_\mathcal{S}) p(V_g \,|\, U_\mathcal{S})} \right) \quad (11)$$

The above removes the unwanted correlation between generalization and complexity measure that is caused by hyperparameter types in set $\mathcal{S}$. Since in our case the conditional mutual information between a complexity measure and generalization is at most equal to the conditional entropy of generalization, we normalize it with the conditional entropy to arrive at a criterion ranging between 0 and 1:

$$\mathcal{H}(V_g \,|\, U_\mathcal{S}) = -\sum_{U_\mathcal{S}} p(U_\mathcal{S}) \sum_{V_g \in \{\pm 1\}} p(V_g \,|\, U_\mathcal{S}) \log(p(V_g \,|\, U_\mathcal{S})) \quad (12)$$

---

[7]For example, if $\mathcal{S}$ contains a single hyperparameter type such as the learning rate, then the conditional mutual information is conditioned on learning rate being observed.

$$\hat{\mathcal{I}}(V_\mu, V_g \mid U_\mathcal{S}) = \frac{\mathcal{I}(V_\mu, V_g \mid U_\mathcal{S})}{\mathcal{H}(V_g \mid U_\mathcal{S})} \tag{13}$$

According to the IC algorithm, an edge is kept between two nodes if there exists no subset $\mathcal{S}$ of hyperparameter types such that the two nodes are independent, i.e. $\hat{\mathcal{I}}(V_\mu, V_g \mid U_\mathcal{S}) = 0$. In our setup, setting $\mathcal{S}$ to the set of all hyperparameter types is not possible as both the conditional entropy and conditional mutual information would become zero. Moreover, due to computational reasons, we only look at $|\mathcal{S}| \leq 2$:

$$\mathcal{K}(\mu) = \min_{U_\mathcal{S} \text{ s.t } |\mathcal{S}| \leq 2} \hat{\mathcal{I}}(V_\mu, V_g \mid U_\mathcal{S}) \tag{14}$$

At a high level, the larger $\mathcal{K}$ is for a measure $\mu$, the more likely an edge exists between $\mu$ and $g$, and therefore the more likely $\mu$ can explain generalization. For details on the set-up, please refer to Appendix B.1 on how these quantities are estimated.

## B.1  DEFINITION OF RANDOM VARIABLES

Since the measures are results of complicated interactions between the data, the model, and the training procedures, we cannot manipulate it to be any values that we want. Instead, we use the following definition of random variables: suppose $\mathcal{S}$ is a subset of all the components of $\boldsymbol{\theta}$ (e.g. $\mathcal{S} = \{\varnothing\}$ for $|\mathcal{S}| = 0$, $|\mathcal{S}| = \{\text{learning rate}\}$ for $|\mathcal{S}| = 1$ or $|\mathcal{S}| = \{\text{learning rate, dropout}\}$ for $|\mathcal{S}| = 2$ ). Specifically we denote $\mathcal{S}_{ab}$ as the collective condition $\{\theta_1^{(a)} = v_1, \theta_1^{(b)} = v_2, \ldots, \theta_{|\mathcal{S}|}^{(a)} = v_{2|\mathcal{S}|-1}, \theta_{|\mathcal{S}|}^{(b)} = v_{2|\mathcal{S}|}\}$. We can then define and empirical measure four probability $\Pr(\mu^{(a)} > \mu^{(b)}, g^{(a)} > g^{(b)} \mid \mathcal{S}_{ab})$, $\Pr(\mu^{(a)} > \mu^{(b)}, g^{(a)} < g^{(b)} \mid \mathcal{S}_{ab})$, $\Pr(\mu^{(a)} < \mu^{(b)}, g^{(a)} > g^{(b)} \mid \mathcal{S}_{ab})$ and $\Pr(\mu^{(a)} < \mu^{(b)}, g^{(a)} < g^{(b)} \mid \mathcal{S}_{ab})$.

|  | $\mu^{(a)} > \mu^{(b)}$ | $\mu^{(a)} \leq \mu^{(b)}$ |
|---|---|---|
| $g^{(a)} > g^{(b)}$ | $p_{00}$ | $p_{01}$ |
| $g^{(a)} \leq g^{(b)}$ | $p_{10}$ | $p_{11}$ |

Figure 3: Joint Probability table for a single $\mathcal{S}_{ab}$

Together forms a 2 by 2 table that defines the joint distribution of the Bernoulli random variables $\Pr(g^{(a)} > g^{(b)} \mid \mathcal{S}_{ab})$ and $\Pr(\mu^{(a)} > \mu^{(b)} \mid \mathcal{S}_{ab})$. For notation convenience, we use $\Pr(\mu, g \mid \mathcal{S}_{ab})$, $\Pr(g \mid \mathcal{S}_{ab})$ and $\Pr(\mu \mid \mathcal{S}_{ab})$ to denote the joint and marginal. If there are $N = 3$ choices for each hyperparameter in $\mathcal{S}$ then there will be $N^{|\mathcal{S}|}$ such tables for each hyperparameter combination. Since each configuration occurs with equal probability, for that arbitrary $\boldsymbol{\theta}^{(a)}$ and $\boldsymbol{\theta}^{(b)}$ drawn from $\Theta$ conditioned on that the components of $\mathcal{S}$ are observed for both models, the joint distribution can be defined as $\Pr(\mu, g \mid \mathcal{S}) = \frac{1}{N^{|\mathcal{S}|}} \sum_{\mathcal{S}_{ab}} \Pr(\mu, g \mid \mathcal{S}_{ab})$ and likely the marginals can be defined as $\Pr(\mu \mid \mathcal{S}) = \frac{1}{N^{|\mathcal{S}|}} \sum_{\mathcal{S}_{ab}} \Pr(\mu \mid \mathcal{S}_{ab})$ and $\Pr(g \mid \mathcal{S}) = \frac{1}{N^{|\mathcal{S}|}} \sum_{\mathcal{S}_{ab}} \Pr(g \mid \mathcal{S}_{ab})$. With these notations established, all the relevant quantities can be computed by iterating over all pairs of models.

## C  EXPERIMENTS

### C.1  MORE TRAINING DETAILS

During our experiments, we found that Batch Normalization (Ioffe and Szegedy, 2015) is crucial to reliably reach a low cross-entropy value for all models; since normalization is a indispensable components of modern neural networks, we decide to use batch normalization in all of our models. We *remove* batch normalization before computing any measure by fusing the $\gamma$, $\beta$ and moving statistics with the convolution operator that precedes the normalization. This is important as Dinh et al. (2017) showed that common generalization measures such as sharpness can be easily manipulated with re-parameterization. We also discovered that the models trained with data augmentation often cannot fit the data (i.e. reach cross-entropy 0.01) completely. Since a model with data augmentation tends to consistently generalize better than the models without data augmentation, measure that reflects the training error (i.e. value of cross-entropy) will easily predict the ranking between two models

even though it has only learned that one model uses data augmentation (see the thought experiments from the previous section). While certain hyperparameter configuration can reach cross-entropy of 0.01 even with data augmentation, it greatly limits the space of models that we can study. Hence, we make the design choice to not include data augmentation in the models of this study. Note that from a theoretical perspective, data augmentation is also challenging to analyze since the training samples generated from the procedure are no longer identical and independently distributed. All values for all the measures we computed over these models can be found in Table 5 in Appendix C.5.

## C.2    THE CHOICE OF STOPPING CRITERION

The choice of stopping criterion is very essential and could completely change the evaluation and the resulting conclusions. In our experiments we noticed that if we pick the stopping criterion based on number of iterations or number of epochs, then since some models optimize faster than others, they end up fitting the training data more and in that case the cross-entropy itself can be very predictive of generalization. To make it harder to distinguish models based on their training performance, it makes more sense to choose the stopping criterion based on the training error or training loss. We noticed that as expected, models with the same cross-entropy usually have very similar training error so that suggests that this choice is not very important. However, during the optimization the training error behavior is noisier than cross-entropy and moreover, after the training error reaches zero, it cannot distinguish models while the cross-entropy is still meaningful after fitting the data. Therefore, we decided to use cross-entropy as the stopping criterion.

## C.3    ALL MODEL SPECIFICATION

As mentioned in the main text, the models we use resemble Network-in-Network (Gao et al., 2011) which is a class of more parameter efficient convolution neural networks that achieve reasonably competitive performance on modern image classification benchmarks. The model consists blocks of modules that have 1 $3 \times 3$ convolution with stride 2 followed by 2 $1 \times 1$ convolution with stride 1. We refer to this single module as a `NiN-block` and construct models of different size by stacking NiN-block. For simplicity, all NiN-block have the same number of output channels $c_{out}$. Dropout is applied at the end of every NiN-block. At the end of the model, there is a $1 \times 1$ convolution reducing the channel number to the class number (i.e. 10 for CIFAR-10) followed by a global average pooling to produce the output logits.

For width, we choose from $c_{out}$ from 3 options: $\{2 \times 96, 4 \times 96, 8 \times 96\}$.

For depth, we choose from 3 options: $\{2 \times \text{NiNblock}, 4 \times \text{NiNblock}, 8 \times \text{NiNblock}\}$

For dropout, we choose from 3 options: $\{0.0, 0.25, 0.5\}$

For batch size, we choose from: $\{32, 64, 128\}$

Since each optimizer may require different learning rate and in some cases, different regularization, we fine-tuned the hyper-parameters for each optimizer while keeping 3 options for every hyper-parameter choices[8].

**Momentum SGD**: We choose momentum of 0.9 and choose the initial learning rate $\eta$ from $\{0.1, 0.032, 0.01\}$ and regularization coefficient $\lambda$ from $\{0.0, 0.0001, 0.0005\}$. The learning rate decay schedule is $\times 0.1$ at iterations $[60000, 90000]$.

**Adam**: We choose initial learning rate $\eta$ from $\{0.001, 3.2e-4, 1e-4\}$, $\epsilon = 1e-3$ and regularization coefficient $\lambda$ from $\{0.0, 0.0001, 0.0005\}$. The learning rate decay schedule is $\times 0.1$ at iterations $[60000, 90000]$.

**RMSProp**: We choose initial learning rate $\eta$ from $\{0.001, 3.2e-4, 1e-4\}$ and regularization coefficient $\lambda$ from $\{0.0, 0.0001, 0.0003\}$. The learning rate decay schedule is $\times 0.1$ at iterations $[60000, 90000]$.

---

[8]While methods with adaptive methods generally require less tuning, in practice researchers have observed performance gains from tuning the initial learning rate and learning rate decay.

## C.4  CANONICAL MEASURES

Based on empirical observations made by the community as a whole, the canonical ordering we give to each of the hyper-parameter categories are as follows:

1. Batchsize: smaller batchsize leads to smaller generalization gap

2. Depth: deeper network leads to smaller generalization gap

3. Width: wider network leads to smaller generalization gap

4. Dropout: The higher the dropout ($\leq 0.5$) the smaller the generalization gap

5. Weight decay: The higher the weight decay (smaller than the maximum for each optimizer) the smaller the generalization gap

6. Learning rate: The higher the learning rate (smaller than the maximum for each optimizer) the smaller the generalization gap

7. Optimizer: Generalization gap of `Momentum SGD` < Generalization gap of `Adam` < Generalization gap of `RMSProp`

## C.5  ALL RESULTS

Below we present all of the measures we computed and their respective $\tau$ and $\Psi$ on more than 10,000 models we trained and additional plots. Unless stated otherwise, convergence is considered when the loss reaches the value of 0.1.

| | ref | batchsize | dropout | learning rate | depth | optimizer | weight decay | width | overall $\tau$ | $\Psi$ |
|---|---|---|---|---|---|---|---|---|---|---|
| vc dim | 19 | 0.000 | 0.000 | 0.000 | -0.909 | 0.000 | 0.000 | -0.171 | -0.251 | -0.154 |
| # params | 20 | 0.000 | 0.000 | 0.000 | -0.909 | 0.000 | 0.000 | -0.171 | -0.175 | -0.154 |
| sharpness | 51 | 0.537 | -0.523 | 0.449 | 0.826 | 0.221 | 0.233 | -0.004 | 0.282 | 0.248 |
| pacbayes | 48 | 0.372 | -0.457 | 0.042 | 0.644 | 0.179 | -0.179 | -0.142 | 0.064 | 0.066 |
| sharpness-orig | 52 | 0.542 | -0.359 | 0.716 | 0.816 | 0.297 | 0.591 | 0.185 | 0.400 | 0.398 |
| pacbayes-orig | 49 | 0.526 | -0.076 | 0.705 | 0.546 | 0.341 | 0.564 | -0.086 | 0.293 | 0.360 |
| frob-distance | 40 | -0.317 | -0.833 | -0.718 | 0.526 | -0.214 | -0.669 | -0.166 | -0.263 | -0.341 |
| spectral-init | 25 | -0.330 | -0.845 | -0.721 | -0.908 | -0.208 | -0.313 | -0.231 | -0.576 | -0.508 |
| spectral-orig | 26 | -0.262 | -0.762 | -0.665 | -0.908 | -0.131 | -0.073 | -0.240 | -0.537 | -0.434 |
| spectral-orig-main | 28 | -0.262 | -0.762 | -0.665 | -0.908 | -0.131 | -0.073 | -0.240 | -0.537 | -0.434 |
| fro/spec | 33 | 0.563 | 0.351 | 0.744 | -0.898 | 0.326 | 0.665 | -0.053 | -0.008 | 0.243 |
| prod-of-spec | 32 | -0.464 | -0.724 | -0.722 | -0.909 | -0.197 | -0.142 | -0.218 | -0.559 | -0.482 |
| prod-of-spec/margin | 31 | -0.308 | -0.782 | -0.702 | -0.907 | -0.166 | -0.148 | -0.179 | -0.570 | -0.456 |
| sum-of-spec | 35 | -0.464 | -0.724 | -0.722 | 0.909 | -0.197 | -0.142 | -0.218 | 0.102 | -0.223 |
| sum-of-spec/margin | 34 | -0.308 | -0.782 | -0.702 | 0.909 | -0.166 | -0.148 | -0.179 | 0.064 | -0.197 |
| spec-dist | 41 | -0.458 | -0.838 | -0.568 | 0.738 | -0.319 | -0.182 | -0.171 | -0.110 | -0.257 |
| prod-of-fro | 37 | 0.440 | -0.199 | 0.538 | -0.909 | 0.321 | 0.731 | -0.101 | -0.297 | 0.117 |
| prod-of-fro/margin | 36 | 0.513 | -0.291 | 0.579 | -0.907 | 0.364 | 0.739 | -0.088 | -0.295 | 0.130 |
| sum-of-fro | 39 | 0.440 | -0.199 | 0.538 | 0.913 | 0.321 | 0.731 | -0.101 | 0.418 | 0.378 |
| sum-of-fro/margin | 38 | 0.520 | -0.369 | 0.598 | 0.882 | 0.380 | 0.738 | -0.080 | 0.391 | 0.381 |
| 1/margin | 22 | -0.312 | 0.593 | -0.234 | -0.758 | -0.223 | 0.211 | -0.125 | -0.124 | -0.121 |
| neg-entropy | 23 | 0.346 | -0.529 | 0.251 | 0.632 | 0.220 | -0.157 | 0.104 | 0.148 | 0.124 |
| path-norm | 44 | 0.363 | -0.190 | 0.216 | 0.925 | 0.272 | 0.195 | 0.178 | 0.370 | 0.280 |
| path-norm/margin | 43 | 0.363 | 0.017 | 0.148 | 0.922 | 0.230 | 0.280 | 0.173 | 0.374 | 0.305 |
| param-norm | 42 | 0.236 | -0.516 | 0.174 | 0.330 | 0.187 | 0.124 | -0.170 | 0.073 | 0.052 |
| fisher-rao | 45 | 0.396 | 0.147 | 0.240 | -0.516 | 0.120 | 0.551 | 0.177 | 0.090 | 0.160 |
| cross-entropy | 21 | 0.440 | -0.402 | 0.140 | 0.390 | 0.149 | 0.232 | 0.080 | 0.149 | 0.147 |
| $1/\sigma$ pacbayes | 53 | 0.501 | -0.033 | 0.744 | 0.200 | 0.346 | 0.609 | 0.056 | 0.303 | 0.346 |
| $1/\sigma$ sharpness | 54 | 0.532 | -0.326 | 0.711 | 0.776 | 0.296 | 0.592 | 0.263 | 0.399 | 0.406 |
| num-step-0.1-to-0.01-loss | 64 | -0.151 | -0.069 | -0.014 | 0.114 | 0.072 | -0.046 | -0.021 | -0.088 | -0.016 |
| num-step-to-0.1-loss | 63 | -0.664 | -0.861 | -0.255 | 0.440 | -0.030 | -0.628 | 0.043 | -0.264 | -0.279 |
| $1/\alpha'$ sharpness mag | 62 | 0.570 | 0.148 | 0.762 | 0.824 | 0.297 | 0.741 | 0.269 | 0.484 | 0.516 |
| $1/\sigma'$ pacbayes mag | 61 | 0.490 | -0.215 | 0.505 | 0.896 | 0.186 | 0.147 | 0.195 | 0.365 | 0.315 |
| pac-sharpness-mag-init | 59 | -0.293 | -0.841 | -0.698 | -0.909 | -0.240 | -0.631 | -0.171 | -0.225 | -0.541 |
| pac-sharpness-mag-orig | 60 | 0.401 | -0.514 | 0.321 | -0.909 | 0.181 | 0.281 | -0.171 | -0.158 | -0.059 |
| pacbayes-mag-init | 56 | 0.425 | -0.658 | -0.035 | 0.874 | 0.099 | -0.407 | 0.069 | 0.175 | 0.052 |
| pacbayes-mag-orig | 57 | 0.532 | -0.480 | 0.508 | 0.902 | 0.188 | 0.155 | 0.186 | 0.410 | 0.284 |
| grad-noise-final | 66 | 0.452 | 0.119 | 0.427 | 0.141 | 0.245 | 0.432 | 0.230 | 0.311 | 0.292 |
| grad-noise-epoch-1 | 65 | 0.071 | 0.378 | 0.376 | -0.517 | 0.121 | 0.221 | 0.037 | 0.070 | 0.098 |
| oracle 0.01 | | 0.579 | 0.885 | 0.736 | 0.920 | 0.529 | 0.622 | 0.502 | 0.851 | 0.682 |
| oracle 0.02 | | 0.414 | 0.673 | 0.548 | 0.742 | 0.346 | 0.447 | 0.316 | 0.726 | 0.498 |
| oracle 0.05 | | 0.123 | 0.350 | 0.305 | 0.401 | 0.132 | 0.201 | 0.142 | 0.456 | 0.236 |
| oracle 0.1 | | 0.069 | 0.227 | 0.132 | 0.223 | 0.086 | 0.121 | 0.093 | 0.241 | 0.136 |
| canonical ordering | | -0.652 | 0.969 | 0.733 | 0.909 | -0.055 | 0.735 | 0.171 | 0.005 | 0.402 |
| canonical ordering depth | | -0.032 | 0.001 | 0.033 | -0.909 | -0.061 | -0.020 | 0.024 | -0.363 | -0.138 |

Table 5: Complexity measures (rows), hyperparameters (columns) and the **rank-correlation coefficients** with models trained on **CIFAR-10**.

| | batchsize | dropout | learning rate | num_block | optimizer | weight decay | width | $\|\mathcal{S}\|=0$ | $\|\mathcal{S}\|=1$ | $\|\mathcal{S}\|=2$ |
|---|---|---|---|---|---|---|---|---|---|---|
| #-param | 0.0202 | 0.0278 | 0.0259 | 0.0044 | 0.0208 | 0.0216 | 0.0379 | 0.0200 | 0.0036 | 0.0000 |
| -entropy | 0.0120 | 0.0656 | 0.0113 | 0.0086 | 0.0120 | 0.0155 | 0.0125 | 0.0117 | 0.0072 | 0.0065 |
| 1-over-sigma-pacbayes-mag | 0.0884 | 0.1514 | 0.0813 | 0.0399 | 0.1004 | 0.1025 | 0.0986 | 0.0960 | 0.0331 | 0.0241 |
| 1-over-sigma-pacbayes | 0.0661 | 0.1078 | 0.0487 | 0.0809 | 0.0711 | 0.0589 | 0.0858 | 0.0664 | 0.0454 | 0.0340 |
| 1-over-sigma-sharpness-mag | 0.1640 | 0.2572 | 0.1228 | 0.1424 | 0.1779 | 0.1562 | 0.1786 | 0.1741 | 0.1145 | 0.0544 |
| 1-over-sigma-sharpness | 0.1086 | 0.2223 | 0.0792 | 0.0713 | 0.1196 | 0.1041 | 0.1171 | 0.1159 | 0.0592 | 0.0256 |
| cross-entropy | 0.0233 | 0.0850 | 0.0118 | 0.0075 | 0.0159 | 0.0119 | 0.0183 | 0.0161 | 0.0062 | 0.0040 |
| displacement | 0.0462 | 0.0530 | 0.0196 | 0.1559 | 0.0502 | 0.0379 | 0.0506 | 0.0504 | 0.0183 | 0.0128 |
| fisher-rao | 0.0061 | 0.0072 | 0.0020 | 0.0713 | 0.0057 | 0.0014 | 0.0071 | 0.0059 | 0.0013 | 0.0018 |
| fro-over-spec | 0.0019 | 0.0065 | 0.0298 | 0.0777 | 0.0036 | 0.0015 | 0.0005 | 0.0000 | 0.0005 | 0.0013 |
| frob-distance | 0.0462 | 0.0530 | 0.0196 | 0.1559 | 0.0502 | 0.0379 | 0.0506 | 0.0504 | 0.0183 | 0.0128 |
| grad-noise-epoch-1 | 0.0051 | 0.0016 | 0.0028 | 0.0633 | 0.0113 | 0.0027 | 0.0052 | 0.0036 | 0.0013 | 0.0013 |
| grad-noise-final | 0.0623 | 0.0969 | 0.0473 | 0.0934 | 0.0745 | 0.0577 | 0.0763 | 0.0712 | 0.0441 | 0.0329 |
| input-grad-norm | 0.0914 | 0.1374 | 0.1203 | 0.0749 | 0.1084 | 0.0853 | 0.1057 | 0.1042 | 0.0623 | 0.0426 |
| margin | 0.0105 | 0.0750 | 0.0078 | 0.0133 | 0.0108 | 0.0183 | 0.0119 | 0.0108 | 0.0072 | 0.0051 |
| oracle-0.01 | 0.6133 | 0.5671 | 0.6007 | 0.5690 | 0.6171 | 0.6108 | 0.6191 | 0.6186 | 0.4727 | 0.2879 |
| oracle-0.02 | 0.4077 | 0.3557 | 0.3929 | 0.3612 | 0.4124 | 0.4057 | 0.4154 | 0.4130 | 0.2987 | 0.1637 |
| oracle-0.05 | 0.1475 | 0.1167 | 0.1369 | 0.1241 | 0.1515 | 0.1469 | 0.1535 | 0.1515 | 0.0980 | 0.0503 |
| pacbayes-mag-init | 0.0216 | 0.0238 | 0.0274 | 0.0046 | 0.0222 | 0.0210 | 0.0345 | 0.0202 | 0.0038 | 0.0004 |
| pacbayes-mag-orig | 0.1160 | 0.2249 | 0.1006 | 0.0426 | 0.1305 | 0.1316 | 0.1246 | 0.1252 | 0.0354 | 0.0221 |
| pacbayes-orig | 0.0620 | 0.1071 | 0.0392 | 0.0597 | 0.0645 | 0.0550 | 0.0977 | 0.0629 | 0.0365 | 0.0225 |
| pacbayes | 0.0053 | 0.0164 | 0.0084 | 0.0086 | 0.0036 | 0.0066 | 0.0185 | 0.0030 | 0.0036 | 0.0040 |
| parameter-norm | 0.0039 | 0.0197 | 0.0066 | 0.0115 | 0.0064 | 0.0049 | 0.0167 | 0.0039 | 0.0038 | 0.0047 |
| path-norm-over-margin | 0.0943 | 0.1493 | 0.1173 | 0.0217 | 0.1025 | 0.1054 | 0.1090 | 0.1011 | 0.0181 | 0.0139 |
| path-norm | 0.1027 | 0.1230 | 0.1308 | 0.0315 | 0.1056 | 0.1028 | 0.1160 | 0.1030 | 0.0261 | 0.0240 |
| prod-of-spec-over-margin | 0.2466 | 0.3139 | 0.2179 | 0.1145 | 0.2473 | 0.2540 | 0.2497 | 0.2481 | 0.0951 | 0.0483 |
| prod-of-spec | 0.2334 | 0.3198 | 0.2070 | 0.1037 | 0.2376 | 0.2470 | 0.2394 | 0.2385 | 0.0862 | 0.0415 |
| random | 0.0005 | 0.0002 | 0.0005 | 0.0002 | 0.0003 | 0.0006 | 0.0009 | 0.0003 | 0.0001 | 0.0004 |
| sharpness-mag-init | 0.0366 | 0.0460 | 0.0391 | 0.0191 | 0.0374 | 0.0373 | 0.0761 | 0.0368 | 0.0159 | 0.0134 |
| sharpness-mag-orig | 0.0125 | 0.0143 | 0.0195 | 0.0043 | 0.0120 | 0.0134 | 0.0142 | 0.0111 | 0.0036 | 0.0033 |
| sharpness-orig | 0.1117 | 0.2353 | 0.0809 | 0.0658 | 0.1223 | 0.1071 | 0.1254 | 0.1189 | 0.0547 | 0.0224 |
| sharpness | 0.0545 | 0.1596 | 0.0497 | 0.0156 | 0.0586 | 0.0599 | 0.0700 | 0.0583 | 0.0130 | 0.0123 |
| spec-init | 0.2536 | 0.3161 | 0.2295 | 0.1179 | 0.2532 | 0.2584 | 0.2540 | 0.2539 | 0.0980 | 0.0559 |
| spec-orig-main | 0.2266 | 0.2903 | 0.2072 | 0.0890 | 0.2255 | 0.2355 | 0.2262 | 0.2262 | 0.0739 | 0.0382 |
| spec-orig | 0.2197 | 0.2815 | 0.2045 | 0.0808 | 0.2180 | 0.2285 | 0.2181 | 0.2188 | 0.0671 | 0.0359 |
| step-0.1-to-0.01 | 0.0125 | 0.0031 | 0.0055 | 0.0093 | 0.0074 | 0.0043 | 0.0070 | 0.0055 | 0.0026 | 0.0032 |
| step-to-0.1 | 0.0349 | 0.0361 | 0.0397 | 0.1046 | 0.0485 | 0.0380 | 0.0568 | 0.0502 | 0.0303 | 0.0134 |
| sum-of-fro-over-margin | 0.1200 | 0.2269 | 0.1005 | 0.0440 | 0.1207 | 0.1060 | 0.1645 | 0.1227 | 0.0366 | 0.0110 |
| sum-of-fro-over-sum-of-spec | 0.0258 | 0.0392 | 0.0055 | 0.1111 | 0.0312 | 0.0194 | 0.0355 | 0.0297 | 0.0051 | 0.0027 |
| sum-of-fro | 0.1292 | 0.2286 | 0.1115 | 0.0441 | 0.1281 | 0.1134 | 0.1714 | 0.1300 | 0.0366 | 0.0119 |
| sum-of-spec-over-margin | 0.0089 | 0.0292 | 0.0406 | 0.0951 | 0.0089 | 0.0069 | 0.0054 | 0.0051 | 0.0054 | 0.0072 |
| sum-of-spec | 0.0127 | 0.0324 | 0.0466 | 0.0876 | 0.0117 | 0.0096 | 0.0080 | 0.0076 | 0.0079 | 0.0099 |
| vc-dim | 0.0422 | 0.0564 | 0.0518 | 0.0039 | 0.0422 | 0.0443 | 0.0627 | 0.0412 | 0.0033 | 0.0000 |
| conditional entropy | 0.9836 | 0.8397 | 0.9331 | 0.8308 | 0.9960 | 0.9746 | 0.9977 | N/A | N/A | N/A |

Table 6: Complexity measures (rows), hyperparameters (columns) and the **mutual information** with models trained on **CIFAR-10**.

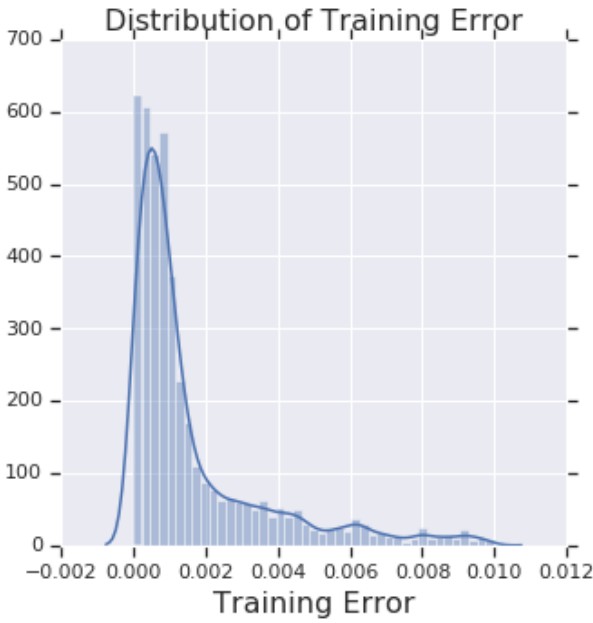

Figure 4: Distribution of training error on the trained models.

| | batchsize | dropout | learning rate | depth | optimizer | weight decay | width | overall $\tau$ | $\Psi$ |
|---|---|---|---|---|---|---|---|---|---|
| vc dim | 0.0000 | 0.0000 | 0.0000 | -1.0000 | 0.0000 | 0.0000 | -0.0478 | -0.3074 | -0.1497 |
| # params | 0.0000 | 0.0000 | 0.0000 | -1.0000 | 0.0000 | 0.0000 | -0.0478 | -0.1934 | -0.1497 |
| sharpness | 0.1898 | -0.4092 | 0.4569 | 0.9752 | 0.1708 | 0.2444 | 0.1202 | 0.5438 | 0.2497 |
| pacbayes | 0.0606 | -0.5806 | 0.0503 | 0.9447 | 0.0831 | -0.2123 | 0.0034 | 0.3688 | 0.0499 |
| sharpness 0ref | 0.2324 | -0.1807 | 0.6329 | 0.9595 | 0.2196 | 0.5018 | 0.1923 | 0.5175 | 0.3654 |
| pacbayes 0ref | 0.1983 | -0.2055 | 0.5979 | 0.8863 | 0.2286 | 0.4583 | 0.0655 | 0.3708 | 0.3185 |
| displacement | -0.1071 | -0.8603 | -0.6270 | 0.8874 | -0.1677 | -0.6319 | -0.0302 | 0.1765 | -0.2196 |
| spectral complexity | -0.2854 | -0.7928 | -0.6423 | -0.9989 | -0.1063 | -0.2913 | -0.0799 | -0.6284 | -0.4567 |
| spectral complexity 0ref | -0.1362 | -0.6110 | -0.4688 | -0.9932 | -0.0513 | 0.0671 | -0.1096 | -0.6163 | -0.3290 |
| spectral complexity 0ref last2 | -0.1362 | -0.6110 | -0.4688 | -0.9628 | -0.0513 | 0.0671 | -0.2797 | -0.5870 | -0.3490 |
| spectral complexity 0ref last1 | 0.6285 | 0.3961 | 0.6646 | -0.8274 | 0.2317 | 0.6047 | 0.0525 | -0.1264 | 0.2501 |
| spectral product | -0.2603 | -0.5835 | -0.6095 | -0.9628 | -0.1063 | -0.0343 | -0.2705 | -0.5615 | -0.4039 |
| spectral product om | -0.2582 | -0.6419 | -0.5852 | -0.9289 | -0.0918 | -0.0681 | -0.2477 | -0.5404 | -0.4031 |
| spectral product dd/2 | -0.2603 | -0.5835 | -0.6095 | 0.9989 | -0.1063 | -0.0343 | -0.2705 | 0.4627 | -0.1237 |
| spectral produce dd/2 om | -0.2582 | -0.6419 | -0.5852 | 0.9921 | -0.0918 | -0.0681 | -0.2477 | 0.4421 | -0.1287 |
| spectral sum | -0.2734 | -0.7752 | -0.3386 | 0.9616 | -0.0669 | -0.2637 | -0.0434 | 0.3542 | -0.1142 |
| frob product | 0.5098 | -0.0369 | 0.5439 | -1.0000 | 0.1861 | 0.6508 | 0.0126 | -0.4983 | 0.1238 |
| frob product om | 0.4673 | -0.1262 | 0.5534 | -1.0000 | 0.2079 | 0.6375 | 0.0091 | -0.5001 | 0.1070 |
| frob product dd/2 | 0.5098 | -0.0369 | 0.5439 | 0.9853 | 0.1861 | 0.6508 | 0.0126 | 0.5928 | 0.4074 |
| frob product dd/2 om | 0.4673 | -0.1262 | 0.5534 | 0.9492 | 0.2079 | 0.6375 | 0.0091 | 0.5638 | 0.3855 |
| median margin | 0.0684 | 0.3861 | -0.1519 | -0.9314 | -0.1018 | 0.3211 | 0.0216 | -0.3829 | -0.0554 |
| input grad norm | 0.0597 | 0.6277 | -0.2289 | 0.9955 | 0.0026 | 0.0383 | 0.0216 | 0.6360 | 0.2166 |
| logit entropy | -0.0320 | -0.4506 | 0.1481 | 0.7999 | 0.1360 | -0.2460 | -0.0106 | 0.3001 | 0.0492 |
| path norm | 0.2150 | 0.2565 | 0.0464 | 0.9854 | 0.1018 | 0.3885 | 0.0614 | 0.5626 | 0.2936 |
| parameter norm | 0.3246 | -0.4794 | 0.1730 | 0.6639 | 0.0780 | 0.1383 | -0.0398 | 0.3747 | 0.1227 |
| fr norm cross-entropy | 0.2313 | 0.0500 | 0.0222 | -0.6189 | 0.1008 | 0.3190 | 0.0546 | -0.2844 | 0.0227 |
| fr norm logit sum | 0.2313 | 0.0500 | 0.0222 | -0.3277 | 0.1008 | 0.3190 | 0.0546 | -0.1168 | 0.0643 |
| fr norm logit margin | 0.2313 | 0.0500 | 0.0222 | -0.3277 | 0.1008 | 0.3190 | 0.0546 | -0.1168 | 0.0643 |
| path norm/margin | 0.1107 | 0.0291 | 0.1340 | 0.9978 | 0.1504 | 0.2098 | 0.0683 | 0.5798 | 0.2429 |
| one epoch loss | 0.4390 | -0.5989 | 0.2624 | 0.9729 | 0.1602 | -0.0445 | -0.0034 | 0.5186 | 0.1697 |
| final loss | 0.0923 | -0.4091 | -0.0042 | -0.0096 | 0.0811 | 0.1118 | -0.0432 | -0.0693 | -0.0258 |
| 1/sigma gaussian | 0.1867 | -0.1862 | 0.6164 | 0.6665 | 0.2280 | 0.4985 | 0.1512 | 0.3148 | 0.3087 |
| 1/sigma sharpness | 0.2321 | -0.1549 | 0.6330 | 0.9363 | 0.2253 | 0.5163 | 0.2179 | 0.4930 | 0.3723 |
| min(norm distance) | 0.3235 | -0.4785 | 0.1727 | 0.6633 | 0.0766 | 0.1391 | -0.0405 | 0.3744 | 0.1223 |
| step between | -0.1224 | -0.1610 | -0.0061 | 0.1556 | 0.0737 | -0.0415 | -0.0154 | -0.0720 | -0.0167 |
| step to | -0.6667 | -0.6982 | -0.4814 | 0.8738 | -0.1609 | -0.6314 | -0.1015 | 0.0035 | -0.2666 |
| step to 0.1 | -0.6656 | -0.9120 | -0.3613 | 0.9556 | -0.1450 | -0.5974 | -0.0414 | 0.0944 | -0.2524 |
| 1/param sharpness | 0.4546 | 0.3254 | 0.6650 | 0.9831 | 0.2753 | 0.6495 | 0.2680 | 0.5676 | 0.5173 |
| 1/param gaussian | 0.2525 | 0.1250 | 0.4758 | 0.9805 | 0.1629 | 0.2698 | 0.0871 | 0.5674 | 0.3362 |
| ratio cplx sharpness | -0.0787 | -0.7181 | -0.4883 | -1.0000 | -0.0640 | -0.4720 | -0.0502 | -0.2254 | -0.4102 |
| ratio cplx sharpness 0ref | 0.5005 | -0.3831 | 0.3153 | -1.0000 | 0.1648 | 0.2440 | -0.0502 | -0.1687 | -0.0298 |
| ratio cplx gaussian | 0.2289 | -0.3322 | 0.2298 | -0.9786 | 0.1625 | -0.0429 | -0.0484 | -0.1309 | -0.1116 |
| ratio cplx gaussian 0ref | 0.0984 | -0.6821 | 0.2351 | -0.9842 | 0.1304 | 0.0542 | -0.0484 | -0.1682 | -0.1709 |
| ratio cplx sharpness u1 | 0.2778 | -0.4237 | 0.5492 | -0.9707 | 0.1830 | 0.4040 | -0.0434 | -0.1580 | -0.0034 |
| ratio cplx sharpness 0ref u1 | 0.3606 | -0.2165 | 0.6476 | -0.9650 | 0.2421 | 0.5463 | -0.0422 | -0.1364 | 0.0818 |
| ratio cplx gaussian u1 | 0.2300 | -0.4279 | -0.0703 | 0.9707 | 0.1346 | -0.3957 | 0.0302 | 0.5052 | 0.0674 |
| ratio cplx gaussian 0ref u1 | 0.4519 | -0.2101 | 0.4876 | 0.9887 | 0.1812 | 0.2924 | 0.1464 | 0.6390 | 0.3340 |
| grad var | 0.2128 | -0.1862 | 0.2458 | 0.0343 | 0.1711 | 0.3211 | 0.1149 | 0.0594 | 0.1305 |
| grad var 1 epoch | 0.1590 | 0.1912 | -0.0159 | 0.0118 | 0.2760 | -0.0046 | 0.1222 | 0.1222 | 0.0806 |
| oracle 0.01 | 0.3811 | 0.6463 | 0.4293 | 0.9517 | 0.3478 | 0.3946 | 0.3572 | 0.8070 | 0.5012 |
| oracle 0.02 | 0.2410 | 0.4102 | 0.2964 | 0.8730 | 0.1886 | 0.2190 | 0.1741 | 0.6854 | 0.3432 |
| oracle 0.05 | 0.1238 | 0.2235 | 0.1530 | 0.6706 | 0.0522 | 0.1057 | 0.0785 | 0.5162 | 0.2010 |
| oracle 0.1 | -0.0239 | 0.0708 | 0.0844 | 0.4356 | 0.0408 | 0.0526 | 0.0512 | 0.3322 | 0.1017 |
| canonical ordering | -0.6732 | 0.9539 | 0.6424 | 1.0000 | -0.1028 | 0.6662 | 0.0478 | 0.0123 | 0.3620 |
| canonical ordering depth | -0.0304 | -0.0247 | 0.0105 | -1.0000 | 0.0253 | -0.0332 | 0.0262 | -0.6241 | |

Table 7: Complexity measures (rows), hyperparameters (columns) and the **rank-correlation coefficients** with models trained on **SVHN** dataset.

| | batchsize | dropout | learning rate | depth | optimizer | weight decay | width | overall $\tau$ | $\Psi$ |
|---|---|---|---|---|---|---|---|---|---|
| vc dim | 0.0000 | 0.0000 | 0.0000 | -0.7520 | 0.0000 | 0.0000 | -0.0392 | -0.1770 | -0.1130 |
| # params | 0.0000 | 0.0000 | 0.0000 | -0.7520 | 0.0000 | 0.0000 | -0.0392 | -0.1194 | -0.1130 |
| sharpness | 0.2059 | -0.1966 | 0.1336 | 0.6358 | -0.0532 | -0.0127 | -0.0317 | 0.2325 | 0.0973 |
| pacbayes | 0.1480 | -0.0488 | -0.0611 | 0.5493 | -0.0570 | -0.2340 | -0.0563 | 0.1477 | 0.0343 |
| sharpness 0ref | 0.2271 | 0.0167 | 0.4462 | 0.6262 | 0.0600 | 0.1563 | -0.0058 | 0.2995 | 0.2181 |
| pacbayes 0ref | 0.2587 | 0.1655 | 0.5282 | 0.5238 | 0.1102 | 0.1318 | -0.0174 | 0.3104 | 0.2430 |
| displacement | -0.1814 | -0.7677 | -0.6504 | 0.3767 | -0.2403 | -0.3831 | -0.0392 | -0.2652 | -0.2693 |
| spectral complexity | -0.1495 | -0.5752 | -0.6208 | -0.7407 | -0.2650 | -0.2885 | -0.0945 | -0.4333 | -0.3906 |
| spectral complexity 0ref | -0.0837 | -0.4196 | -0.4747 | -0.7379 | -0.1776 | -0.1468 | -0.1085 | -0.3860 | -0.3070 |
| spectral complexity 0ref last2 | -0.0837 | -0.4196 | -0.4747 | -0.7284 | -0.1776 | -0.1468 | -0.1857 | -0.3940 | -0.3166 |
| spectral complexity 0ref last1 | 0.2606 | 0.3893 | 0.7221 | -0.7435 | 0.4169 | 0.4404 | 0.0615 | 0.0477 | 0.2210 |
| spectral product | -0.2034 | -0.5619 | -0.6199 | -0.7520 | -0.2184 | -0.1269 | -0.0691 | -0.4176 | -0.3645 |
| spectral product om | -0.1257 | -0.4727 | -0.5549 | -0.7181 | -0.2260 | -0.2113 | -0.1707 | -0.4238 | -0.3542 |
| spectral product dd/2 | -0.2034 | -0.5619 | -0.6199 | 0.7520 | -0.2184 | -0.1269 | -0.0691 | 0.0547 | -0.1496 |
| spectral produce dd/2 om | -0.1257 | -0.4727 | -0.5549 | 0.7501 | -0.2260 | -0.2113 | -0.1707 | 0.0868 | -0.1445 |
| spectral sum | -0.2005 | -0.8378 | -0.5692 | 0.5832 | -0.3751 | -0.0899 | -0.0392 | -0.1517 | -0.2184 |
| frob product | 0.2854 | -0.1532 | 0.4967 | -0.7520 | 0.3609 | 0.4656 | 0.0054 | -0.2162 | 0.1013 |
| frob product om | 0.2816 | -0.1987 | 0.4613 | -0.7520 | 0.2365 | 0.3729 | 0.0130 | -0.2113 | 0.0592 |
| frob product dd/2 | 0.2854 | -0.1532 | 0.4967 | 0.7652 | 0.3609 | 0.4656 | 0.0054 | 0.3407 | 0.3180 |
| frob product dd/2 om | 0.2816 | -0.1987 | 0.4613 | 0.7643 | 0.2365 | 0.3729 | 0.0130 | 0.3356 | 0.2758 |
| median margin | -0.1652 | 0.3153 | -0.0850 | -0.5474 | 0.1263 | 0.1738 | 0.2142 | -0.1295 | 0.0046 |
| input grad norm | 0.0851 | 0.6548 | -0.2502 | 0.7379 | -0.1871 | -0.0009 | 0.0088 | 0.3563 | 0.1498 |
| logit entropy | 0.2200 | -0.3496 | 0.3906 | 0.5584 | 0.1614 | -0.2819 | -0.2095 | 0.1378 | 0.0699 |
| path norm | 0.2549 | 0.5258 | 0.2951 | 0.8161 | 0.2593 | 0.2223 | 0.0420 | 0.3892 | 0.3451 |
| parameter norm | 0.2472 | -0.0090 | 0.3754 | 0.1287 | 0.2716 | 0.1569 | -0.0458 | 0.0865 | 0.1607 |
| fr norm cross-entropy | 0.0727 | 0.3722 | 0.0162 | -0.5314 | -0.1595 | 0.0355 | 0.0231 | 0.0246 | -0.0245 |
| fr norm logit sum | 0.0727 | 0.3722 | 0.0162 | -0.0844 | -0.1595 | 0.0355 | 0.0231 | 0.1780 | 0.0394 |
| fr norm logit margin | 0.0727 | 0.3722 | 0.0162 | -0.0844 | -0.1595 | 0.0355 | 0.0231 | 0.1780 | 0.0394 |
| path norm/margin | 0.2510 | 0.0441 | 0.3314 | 0.7718 | 0.1206 | 0.0571 | -0.0558 | 0.3580 | 0.2172 |
| one epoch loss | 0.1843 | -0.4509 | 0.0544 | 0.0655 | 0.0684 | -0.0012 | -0.0425 | -0.1217 | -0.0174 |
| final loss | 0.1452 | -0.1095 | -0.0630 | 0.3484 | -0.2080 | -0.1140 | -0.2236 | 0.1410 | -0.0321 |
| 1/sigma gaussian | 0.2525 | 0.1905 | 0.4993 | 0.3698 | 0.0822 | 0.1298 | 0.0660 | 0.3213 | 0.2272 |
| 1/sigma sharpness | 0.2120 | 0.0008 | 0.3879 | 0.6097 | 0.0161 | 0.1191 | -0.0073 | 0.3005 | 0.1912 |
| min(norm distance) | 0.2472 | -0.0090 | 0.3754 | 0.1287 | 0.2716 | 0.1569 | -0.0458 | 0.0865 | 0.1607 |
| step between | -0.0053 | -0.0747 | -0.0792 | 0.1688 | 0.0318 | 0.0621 | -0.0168 | -0.0210 | 0.0124 |
| step to | -0.3219 | -0.5252 | -0.4186 | 0.3199 | -0.1076 | -0.4497 | -0.0095 | -0.2071 | -0.2161 |
| step to 0.1 | -0.3219 | -0.8336 | -0.2626 | 0.2859 | -0.0699 | -0.4231 | -0.0062 | -0.2350 | -0.2331 |
| 1/param sharpness | 0.2127 | 0.2602 | 0.4458 | 0.6430 | 0.0354 | 0.1846 | 0.0071 | 0.3613 | 0.2555 |
| 1/param gaussian | 0.1660 | 0.0065 | 0.4001 | 0.6820 | 0.0319 | -0.0879 | -0.1308 | 0.2878 | 0.1525 |
| ratio cplx sharpness | -0.1776 | -0.7743 | -0.6476 | -0.7520 | -0.2498 | -0.3803 | -0.0392 | -0.1602 | -0.4315 |
| ratio cplx sharpness 0ref | 0.3789 | -0.0109 | 0.5033 | -0.7520 | 0.3067 | 0.2688 | -0.0392 | -0.0867 | 0.0937 |
| ratio cplx gaussian | 0.1404 | -0.2537 | 0.1203 | -0.7501 | 0.0446 | -0.2183 | -0.0392 | -0.1123 | -0.1366 |
| ratio cplx gaussian 0ref | 0.1309 | -0.4026 | 0.2961 | -0.7520 | 0.0389 | -0.1434 | -0.0392 | -0.1075 | -0.1245 |
| ratio cplx sharpness u1 | 0.2091 | -0.1873 | 0.1958 | -0.7520 | -0.0114 | 0.1140 | -0.0392 | -0.0971 | -0.0673 |
| ratio cplx sharpness 0ref u1 | 0.2615 | 0.0669 | 0.5110 | -0.7520 | 0.1652 | 0.2527 | -0.0392 | -0.0774 | 0.0666 |
| ratio cplx gaussian u1 | 0.0658 | -0.2413 | -0.0411 | 0.6690 | 0.0047 | -0.3558 | -0.1296 | 0.1672 | -0.0040 |
| ratio cplx gaussian 0ref u1 | 0.2234 | -0.0346 | 0.4737 | 0.6954 | 0.0722 | -0.0239 | -0.0468 | 0.3329 | 0.1942 |
| grad var | 0.1013 | 0.3514 | 0.3706 | 0.2730 | 0.1035 | -0.0652 | 0.0250 | 0.3538 | 0.1656 |
| grad var 1 epoch | 0.0801 | 0.4045 | 0.3792 | -0.3701 | 0.1349 | 0.1328 | 0.0814 | 0.1279 | 0.1204 |
| oracle 0.01 | 0.5789 | 0.8862 | 0.7507 | 0.8274 | 0.5878 | 0.5464 | 0.5123 | 0.8470 | 0.6700 |
| oracle 0.02 | 0.3588 | 0.7288 | 0.5922 | 0.5804 | 0.3970 | 0.3440 | 0.3927 | 0.7032 | 0.4848 |
| oracle 0.05 | 0.1114 | 0.4149 | 0.3066 | 0.2937 | 0.1918 | 0.1473 | 0.1697 | 0.4267 | 0.2336 |
| oracle 0.1 | 0.1037 | 0.2281 | 0.1738 | 0.1957 | 0.1225 | 0.0692 | 0.0876 | 0.2423 | 0.1401 |
| canonical ordering | -0.3254 | 0.9459 | 0.7125 | 0.7520 | -0.0598 | 0.4628 | 0.0392 | -0.0151 | 0.3610 |
| canonical ordering depth | -0.0238 | -0.0337 | 0.0105 | -0.7520 | -0.0152 | 0.0353 | -0.0054 | -0.2835 | -0.1120 |

Table 8: Complexity measures (rows), hyperparameters (columns) and the **rank-correlation coefficients** with models trained on **CIFAR-10** when converged to **Loss = 0.1**.

| | batchsize | dropout | learning rate | depth | optimizer | weight decay | width | overall $\tau$ | $\Phi$ |
|---|---|---|---|---|---|---|---|---|---|
| vc dim | 0 | 0 | 0 | -0.9073 | 0 | 0 | -0.1487 | -0.2509 | -0.1509 |
| # params | 0 | 0 | 0 | -0.9073 | 0 | 0 | -0.1487 | -0.1751 | -0.1509 |
| sharpness | 0.5492 | -0.5155 | 0.4636 | 0.8247 | 0.2134 | 0.2025 | 0.0083 | 0.2848 | 0.2495 |
| pacbayes | 0.3896 | -0.4459 | 0.0427 | 0.6289 | 0.1721 | -0.1757 | -0.1266 | 0.0647 | 0.0693 |
| sharpness-orig | 0.5493 | -0.3492 | 0.7147 | 0.8101 | 0.3006 | 0.5655 | 0.1976 | 0.3996 | 0.3984 |
| pacbayes-orig | 0.5399 | -0.0847 | 0.7237 | 0.5377 | 0.3561 | 0.5597 | -0.0693 | 0.2895 | 0.3662 |
| frob-distance | -0.3048 | -0.8366 | -0.7253 | 0.5301 | -0.2437 | -0.6701 | -0.1499 | -0.2606 | -0.3429 |
| spec-init | -0.3414 | -0.8436 | -0.7326 | -0.9068 | -0.2422 | -0.3134 | -0.2133 | -0.5743 | -0.5133 |
| spec-orig | -0.2633 | -0.7593 | -0.678 | -0.9068 | -0.1611 | -0.0683 | -0.2273 | -0.5354 | -0.4377 |
| spec-orig-main | -0.2633 | -0.7593 | -0.678 | -0.9064 | -0.1611 | -0.0683 | -0.2662 | -0.5451 | -0.4432 |
| fro / spec | 0.5884 | 0.3703 | 0.7501 | -0.9014 | 0.3661 | 0.658 | -0.0219 | -0.0086 | 0.2585 |
| prod-of-spec | -0.4718 | -0.7237 | -0.7302 | -0.9072 | -0.2385 | -0.1409 | -0.2126 | -0.5598 | -0.4893 |
| prod-of-spec/margin | -0.3222 | -0.7803 | -0.716 | -0.9066 | -0.2066 | -0.1614 | -0.1727 | -0.5698 | -0.4665 |
| sum-of-spec | -0.4718 | -0.7237 | -0.7302 | 0.9072 | -0.2385 | -0.1409 | -0.2126 | 0.1023 | -0.2301 |
| sum-of-spec/margin | -0.3222 | -0.7803 | -0.716 | 0.9066 | -0.2066 | -0.1614 | -0.1727 | 0.0662 | -0.2075 |
| spec-dist | -0.4506 | -0.8263 | -0.5791 | 0.7297 | -0.3413 | -0.2027 | -0.1485 | -0.1044 | -0.2598 |
| prod-of-fro | 0.4659 | -0.1885 | 0.5283 | -0.9072 | 0.3342 | 0.7255 | -0.0835 | -0.2972 | 0.1250 |
| prod-of-fro/margin | 0.5377 | -0.372 | 0.5888 | -0.9072 | 0.4024 | 0.7329 | -0.0673 | -0.2957 | 0.1308 |
| sum-of-fro | 0.4659 | -0.1885 | 0.5283 | 0.9099 | 0.3342 | 0.7255 | -0.0835 | 0.4157 | 0.3845 |
| sum-of-fro/margin | 0.5377 | -0.372 | 0.5888 | 0.8832 | 0.4024 | 0.7329 | -0.0673 | 0.3894 | 0.3865 |
| 1/margin | -0.3334 | 0.5914 | -0.2543 | -0.7539 | -0.2257 | 0.2097 | -0.0988 | -0.1257 | -0.1236 |
| input grad norm | 0.5235 | 0.263 | 0.0544 | 0.6239 | 0.0888 | 0.5969 | 0.2054 | 0.3836 | 0.3366 |
| neg-entropy | 0.3686 | -0.5443 | 0.2609 | 0.6326 | 0.2296 | -0.1567 | 0.0973 | 0.1472 | 0.1269 |
| path-norm | 0.2457 | 0.262 | 0.0397 | 0.9296 | 0.1271 | 0.3291 | 0.1558 | 0.3718 | 0.2984 |
| param-norm | 0.2414 | -0.5194 | 0.1611 | 0.3346 | 0.1866 | 0.1198 | -0.1509 | 0.0729 | 0.0533 |
| fisher-rao | 0.4327 | 0.1625 | 0.2494 | -0.5317 | 0.1322 | 0.5559 | 0.1484 | 0.1028 | 0.1642 |
| fr norm logit sum | 0.4327 | 0.1625 | 0.2494 | -0.094 | 0.1322 | 0.5559 | 0.1484 | 0.2238 | 0.2267 |
| fr norm logit margin | 0.4327 | 0.1625 | 0.2494 | -0.094 | 0.1322 | 0.5559 | 0.1484 | 0.2238 | 0.2267 |
| path norm/margin | 0.3692 | -0.2022 | 0.2159 | 0.9189 | 0.2523 | 0.2103 | 0.1582 | 0.3724 | 0.2747 |
| one epoch loss | 0.3939 | -0.4362 | 0.0477 | 0.1573 | 0.1149 | -0.0475 | 0.0128 | -0.0147 | 0.0347 |
| cross-entropy | 0.4443 | -0.4015 | 0.1518 | 0.3821 | 0.1367 | 0.2322 | 0.0676 | 0.1515 | 0.1447 |
| 1/sigma pacbayes | 0.5109 | -0.0349 | 0.7551 | 0.2032 | 0.3738 | 0.6048 | 0.0686 | 0.2993 | 0.3545 |
| 1/sigma sharpness | 0.536 | -0.3169 | 0.7154 | 0.7529 | 0.3021 | 0.5726 | 0.3976 | 0.3976 | 0.4034 |
| min(norm distance) | 0.2414 | -0.5194 | 0.1611 | 0.3346 | 0.1866 | 0.1198 | -0.1509 | 0.0729 | 0.0533 |
| num-step-0.1-to-0.01-loss | -0.1458 | -0.0816 | -0.0166 | 0.1318 | 0.0949 | -0.0348 | -0.0387 | -0.086 | -0.0130 |
| step to | -0.6798 | -0.5418 | -0.4441 | 0.3493 | -0.0578 | -0.6909 | 0.0102 | -0.2812 | -0.2936 |
| num-step-to-0.1-loss | -0.68 | -0.8526 | -0.2662 | 0.4545 | -0.0291 | -0.6484 | 0.0291 | -0.2626 | -0.2847 |
| 1/alpha sharpness mag | 0.5802 | 0.1381 | 0.7537 | 0.8181 | 0.3163 | 0.7371 | 0.2416 | 0.481 | 0.5122 |
| 1/alpha pacbayes mag | 0.5089 | -0.2388 | 0.5203 | 0.8959 | 0.1907 | 0.1628 | 0.1738 | 0.3649 | 0.3162 |
| pac-sharpness-mag-init | -0.2967 | -0.8451 | -0.7165 | -0.9072 | -0.2637 | -0.6387 | -0.1488 | -0.2256 | -0.5452 |
| pac-sharpness-mag-orig | 0.4145 | -0.5227 | 0.3102 | -0.9072 | 0.1916 | 0.2586 | -0.1488 | -0.159 | -0.0577 |
| pacbayes-mag-init | 0.4783 | -0.6438 | 0.2402 | -0.9072 | 0.1446 | -0.1006 | -0.1488 | -0.1669 | -0.1339 |
| pacbayes-mag-orig | 0.4694 | -0.7749 | 0.317 | -0.9072 | 0.1343 | 0.0315 | -0.1488 | -0.1682 | -0.1255 |
| ratio cplx sharpness u1 | 0.5034 | -0.5539 | 0.6314 | -0.9064 | 0.2799 | 0.4205 | -0.1487 | -0.1424 | 0.0323 |
| ratio cplx sharpness 0ref u1 | 0.5602 | -0.3762 | 0.7642 | -0.9062 | 0.3653 | 0.6861 | -0.1487 | -0.1237 | 0.1350 |
| ratio cplx gaussian u1 | 0.4365 | -0.6655 | -0.0286 | 0.8761 | 0.1058 | -0.403 | 0.0465 | 0.1778 | 0.0525 |
| ratio cplx gaussian 0ref u1 | 0.5721 | -0.4788 | 0.5105 | 0.9018 | 0.1896 | 0.1495 | 0.168 | 0.4093 | 0.2875 |
| grad-noise-final | 0.3663 | 0.0039 | 0.3066 | 0.0813 | 0.1773 | 0.4492 | 0.2615 | 0.2521 | 0.2209 |
| grad-noise-epoch-1 | -0.0376 | 0.3618 | 0.2691 | -0.5688 | -0.0342 | 0.2535 | -0.0616 | -0.0252 | 0.0260 |
| oracle 0.01 | 0.588 | 0.8718 | 0.7047 | 0.9094 | 0.5191 | 0.6117 | 0.5107 | 0.852 | 0.6736 |
| oracle 0.02 | 0.3904 | 0.6862 | 0.5405 | 0.7226 | 0.35 | 0.3969 | 0.336 | 0.7197 | 0.4889 |
| oracle 0.05 | 0.1827 | 0.3694 | 0.3099 | 0.3893 | 0.1478 | 0.1676 | 0.1665 | 0.4518 | 0.2476 |
| oracle 0.1 | 0.106 | 0.2132 | 0.1694 | 0.2084 | 0.0922 | 0.0859 | 0.082 | 0.259 | 0.1367 |
| canonical ordering | -0.668 | 0.9753 | 0.7421 | 0.9073 | -0.0511 | 0.7268 | 0.1487 | -0.0039 | 0.3973 |
| canonical ordering depth | 0.0025 | -0.012 | -0.0019 | -0.9073 | 0.0041 | -0.0133 | -0.0002 | -0.3605 | -0.1326 |

Table 9: Complexity measures (rows), hyperparameters (columns) and the **average rank-correlation coefficients over 5 runs** with models trained on **CIFAR-10**. The numerical values are consistent of that of Table 5.

| | batchsize | dropout | learning rate | depth | optimizer | weight decay | width | overall $\tau$ | $\Psi$ |
|---|---|---|---|---|---|---|---|---|---|
| vc dim | 0 | 0 | 0 | 0.0038 | 0 | 0 | 0.0179 | 0.0006 | 0.0026 |
| # params | 0 | 0 | 0 | 0.0038 | 0 | 0 | 0.0179 | 0.0009 | 0.0026 |
| sharpness | 0.0124 | 0.0129 | 0.0153 | 0.0036 | 0.0196 | 0.0154 | 0.0181 | 0.0026 | 0.0056 |
| pacbayes | 0.0171 | 0.0159 | 0.0108 | 0.0086 | 0.0074 | 0.0078 | 0.0169 | 0.0008 | 0.0048 |
| sharpness-orig | 0.0082 | 0.0106 | 0.0062 | 0.0073 | 0.0192 | 0.0151 | 0.0164 | 0.0034 | 0.0048 |
| pacbayes-orig | 0.011 | 0.0062 | 0.0111 | 0.0083 | 0.0162 | 0.013 | 0.0173 | 0.0025 | 0.0047 |
| frob-distance | 0.0102 | 0.0049 | 0.0067 | 0.0058 | 0.017 | 0.0102 | 0.0176 | 0.0035 | 0.0043 |
| spec-init | 0.0061 | 0.0029 | 0.0072 | 0.004 | 0.0192 | 0.0191 | 0.0127 | 0.001 | 0.0045 |
| spec-orig | 0.0015 | 0.0096 | 0.0072 | 0.004 | 0.0166 | 0.0234 | 0.0136 | 0.0009 | 0.0049 |
| spec-orig-main | 0.0015 | 0.0096 | 0.0072 | 0.0037 | 0.0166 | 0.0234 | 0.0083 | 0.0004 | 0.0046 |
| fro / spec | 0.0164 | 0.0105 | 0.0034 | 0.0048 | 0.0205 | 0.0151 | 0.0203 | 0.0024 | 0.0055 |
| prod-of-spec | 0.0053 | 0.0109 | 0.0048 | 0.0037 | 0.0237 | 0.0249 | 0.0101 | 0.0008 | 0.0055 |
| prod-of-spec/margin | 0.0075 | 0.0078 | 0.0082 | 0.0039 | 0.0225 | 0.0232 | 0.0054 | 0.0006 | 0.0051 |
| sum-of-spec | 0.0053 | 0.0109 | 0.0048 | 0.0037 | 0.0237 | 0.0249 | 0.0101 | 0.0014 | 0.0055 |
| sum-of-spec/margin | 0.0075 | 0.0078 | 0.0082 | 0.0035 | 0.0225 | 0.0232 | 0.0054 | 0.0015 | 0.0051 |
| spec-dist | 0.012 | 0.0095 | 0.0081 | 0.0084 | 0.0221 | 0.0122 | 0.0177 | 0.0036 | 0.0052 |
| prod-of-fro | 0.016 | 0.0096 | 0.0117 | 0.0037 | 0.0191 | 0.0121 | 0.0174 | 0.0014 | 0.0052 |
| prod-of-fro/margin | 0.0112 | 0.0126 | 0.0083 | 0.0037 | 0.0224 | 0.0093 | 0.0141 | 0.0014 | 0.0049 |
| sum-of-fro | 0.016 | 0.0096 | 0.0117 | 0.0034 | 0.0191 | 0.0121 | 0.0174 | 0.0024 | 0.0052 |
| sum-of-fro/margin | 0.0112 | 0.0126 | 0.0083 | 0.0054 | 0.0224 | 0.0093 | 0.0141 | 0.002 | 0.0049 |
| 1/margin | 0.0191 | 0.0059 | 0.0154 | 0.0068 | 0.0221 | 0.0079 | 0.0224 | 0.0026 | 0.0060 |
| input grad norm | 0.0147 | 0.0186 | 0.019 | 0.0018 | 0.0222 | 0.0161 | 0.011 | 0.0043 | 0.0061 |
| neg-entropy | 0.0163 | 0.0169 | 0.012 | 0.0093 | 0.022 | 0.0184 | 0.0204 | 0.0025 | 0.0064 |
| path-norm | 0.0103 | 0.006 | 0.0079 | 0.0034 | 0.0174 | 0.0115 | 0.0178 | 0.0014 | 0.0044 |
| param-norm | 0.0125 | 0.0061 | 0.0071 | 0.0077 | 0.0083 | 0.0051 | 0.0175 | 0.0016 | 0.0038 |
| fisher-rao | 0.0192 | 0.0153 | 0.0084 | 0.0311 | 0.01 | 0.0158 | 0.0069 | 0.0065 | |
| fr norm logit sum | 0.0192 | 0.0153 | 0.0084 | 0.0169 | 0.0311 | 0.01 | 0.0158 | 0.0075 | 0.0068 |
| fr norm logit margin | 0.0192 | 0.0153 | 0.0084 | 0.0169 | 0.0311 | 0.01 | 0.0158 | 0.0075 | 0.0068 |
| path norm/margin | 0.0095 | 0.0172 | 0.0054 | 0.0056 | 0.0157 | 0.0224 | 0.0192 | 0.0019 | 0.0056 |
| one epoch loss | 0.0169 | 0.0128 | 0.0146 | 0.0066 | 0.0223 | 0.0126 | 0.0173 | 0.005 | 0.0058 |
| cross-entropy | 0.0221 | 0.0128 | 0.0174 | 0.0138 | 0.0151 | 0.014 | 0.0183 | 0.0023 | 0.0062 |
| 1/sigma pacbayes | 0.0095 | 0.0031 | 0.0081 | 0.0066 | 0.0173 | 0.0132 | 0.0162 | 0.0035 | 0.0044 |
| 1/sigma sharpness | 0.0084 | 0.009 | 0.0077 | 0.0126 | 0.0185 | 0.0119 | 0.0121 | 0.0039 | 0.0045 |
| min(norm distance) | 0.0125 | 0.0061 | 0.0071 | 0.0077 | 0.0083 | 0.0051 | 0.0175 | 0.0016 | 0.0038 |
| num-step-0.1-to-0.01-loss | 0.0049 | 0.0094 | 0.0071 | 0.0182 | 0.0147 | 0.0081 | 0.0222 | 0.0023 | 0.0051 |
| step to | 0.0118 | 0.011 | 0.0162 | 0.0169 | 0.0135 | 0.0101 | 0.012 | 0.002 | 0.0050 |
| num-step-to-0.1-loss | 0.0119 | 0.0059 | 0.0101 | 0.0236 | 0.0191 | 0.0148 | 0.0152 | 0.002 | 0.0058 |
| 1/alpha sharpness mag | 0.0108 | 0.0224 | 0.0048 | 0.0082 | 0.0262 | 0.0097 | 0.0201 | 0.0031 | 0.0062 |
| 1/alpha pacbayes mag | 0.0198 | 0.0166 | 0.0084 | 0.0037 | 0.0228 | 0.015 | 0.0237 | 0.0044 | 0.0065 |
| pac-sharpness-mag-init | 0.0113 | 0.0039 | 0.0139 | 0.0037 | 0.0186 | 0.0155 | 0.0179 | 0.0011 | 0.0051 |
| pac-sharpness-mag-orig | 0.016 | 0.0061 | 0.0127 | 0.0037 | 0.0188 | 0.0139 | 0.0179 | 0.0008 | 0.0052 |
| pacbayes-mag-init | 0.022 | 0.0059 | 0.0171 | 0.0037 | 0.0173 | 0.0131 | 0.0179 | 0.001 | 0.0057 |
| pacbayes-mag-orig | 0.0221 | 0.0077 | 0.0083 | 0.0037 | 0.0213 | 0.0134 | 0.0179 | 0.0009 | 0.0057 |
| ratio cplx sharpness u1 | 0.0177 | 0.0134 | 0.0127 | 0.0036 | 0.0261 | 0.012 | 0.0183 | 0.0009 | 0.0061 |
| ratio cplx sharpness 0ref u1 | 0.0124 | 0.0079 | 0.0052 | 0.0039 | 0.0266 | 0.0056 | 0.0183 | 0.0006 | 0.0052 |
| ratio cplx gaussian u1 | 0.0205 | 0.0106 | 0.0075 | 0.0019 | 0.0156 | 0.01 | 0.0218 | 0.0031 | 0.0054 |
| ratio cplx gaussian 0ref u1 | 0.0239 | 0.0126 | 0.0035 | 0.0028 | 0.0173 | 0.0087 | 0.017 | 0.0041 | 0.0054 |
| grad-noise-final | 0.0447 | 0.0598 | 0.0628 | 0.0337 | 0.0394 | 0.0243 | 0.0363 | 0.0309 | 0.0170 |
| grad-noise-epoch-1 | 0.0547 | 0.0165 | 0.0542 | 0.0316 | 0.082 | 0.0173 | 0.0514 | 0.0478 | 0.0186 |
| oracle 0.01 | 0.0178 | 0.0078 | 0.0153 | 0.0108 | 0.0189 | 0.0086 | 0.026 | 0.0026 | 0.0061 |
| oracle 0.02 | 0.0133 | 0.0135 | 0.0081 | 0.0138 | 0.0272 | 0.0167 | 0.0058 | 0.0033 | 0.0058 |
| oracle 0.05 | 0.0091 | 0.0249 | 0.0133 | 0.0136 | 0.0171 | 0.015 | 0.0239 | 0.0076 | 0.0066 |
| oracle 0.1 | 0.0188 | 0.0333 | 0.0292 | 0.0145 | 0.0185 | 0.0185 | 0.0321 | 0.0107 | 0.0102 |
| canonical ordering | 0.0111 | 0.004 | 0.0073 | 0.0038 | 0.0185 | 0.0108 | 0.0179 | 0.0027 | 0.0045 |
| canonical ordering depth | 0.018 | 0.0226 | 0.0208 | 0.0038 | 0.0198 | 0.0273 | 0.0202 | 0.0046 | 0.0076 |

Table 10: Complexity measures (rows), hyperparameters (columns) and the **standard deviation of each entry measured over 5 runs** with models trained on **CIFAR-10**. The standard deviation for $\Psi$ is computed assuming that each hyperparamters are independent from each other. We see that all standard deviation are quite small, suggesting the results in of Table 5 are statistically significant.

# D    COMPLEXITY MEASURES

In this section, we look at different complexity measures. When a measure $\mu$ is based on a generalization bound, we chose it so that the following is true with probability 0.99 (we choose the failure probability $\delta$ to be 0.01):

$$L \leq \hat{L} + \sqrt{\frac{\mu}{m}} \tag{15}$$

We also consider measures which do not provably bound the generalization error and evaluate those.

Note that in almost all cases, the canonical ordering given based on some "common" assumptions are positively correlated with the generalization in terms of both $\tau$ and $\Psi$; however, for optimizer, the correlation $\tau$ is close to 0. This implies that the choice of optimizer is only essentially uncorrelated with the generalization gap in the range of models we consider. This ordering helps validate many techniques used by the practioners.

## D.1    VC-DIMENSION BASED MEASURES

We start by restating the theorem in (Bartlett et al., 2019) which provides an upper bound on the VC-dimension of any piece-wise linear network.

**Theorem 1 (Bartlett et al. (2019))** *Let $\mathcal{F}$ be the class of feed-forward networks with a fixed computation graph of depth $d$ and ReLU activations. Let $a_i$ and $q_i$ be the number of activations and parameters in layer $i$. Then VC-dimension of $\mathcal{F}$ can be bounded as follows:*

$$VC(\mathcal{F}) \leq d + \left( \sum_{i=1}^{d} (d-i+1)q_i \right) \log_2 \left( 8e \sum_{i=1}^{d} i a_i \log_2 \left( 4e \sum_{j=1}^{d} j a_j \right) \right)$$

**Theorem 2** *Given a convolutional network $f$, for any $\delta > 0$, with probability $1 - \delta$ over the the training set:*

$$L \leq \hat{L} + 4000 \sqrt{\frac{d \log_2 (6dn)^3 \sum_{i=1}^{d} k_i^2 c_i c_{i-1}}{m}} + \sqrt{\frac{\log(1/\delta)}{m}} \tag{16}$$

**Proof** We simplify the bound in Theorem 1 using a $d'$ to refer to the depth instead of $d$:

$$\text{VC}(\mathcal{F}) \leq d' + \left( \sum_{i=1}^{d'} (d-i+1)q_i \right) \log_2 \left( 8e \sum_{i=1}^{d'} i a_i \log_2 \left( 4e \sum_{j=1}^{d'} j a_j \right) \right)$$

$$\leq d' + \left( \sum_{i=1}^{d'} (d'-i+1)q_i \right) \log_2 \left( 8e \sum_{i=1}^{d'} i a_i \right)^2$$

$$\leq d' + 2 \log_2 \left( 8e \sum_{i=1}^{d'} i a_i \right) \sum_{i=1}^{d'} (d'-i+1)q_i$$

$$\leq 3d' \log_2 \left( 8e \sum_{i=1}^{d'} i a_i \right) \sum_{i=1}^{d'} q_i$$

In order to extend the above bound to a convolutional network, we need to present a pooling layer with ReLU activations. First note that maximum of two inputs can be calculated using two layers with ReLU and linear activations as $\max(x_1, x_2) = x_1 + ReLU(x_2 - x_1)$. Now, since max-pooling at layer $i$ has kernel sizes $k_i'$, we need $\lceil 4 \log_2(k_i') \rceil$ layers to present that but given that the kernel size of the max-pooling layer is at most size of the image, we have

$$\lceil 4 \log_2(k_i') \rceil \leq \lceil 4 \log_2(n^2) \rceil \leq \lceil 8 \log_2(n) \rceil \leq 9 \log_2(n)$$

Therefore, we have $d' \leq 9d\log_2(n)$. The number of activations in any of these layers is at most $n^2 c_i$ since there are at most $n^2$ pairs of neighbor pixels in an $n \times n$ image with $c_i$ channels. We ignore strides when calculating the upper bound since it only reduces number of activations at a few layers and does not change the bound significantly. Using these bounds on $d'$, $a_i$ and $q_i$ the equivalent network, we can bound the VC dimension as follows:

$$\mathrm{VC}(\mathcal{F}) \leq 27d\log_2(n)\log_2\left(8e(9d\log_2(n))^2 n^2\right)(9\log_2(n))\sum_{i=1}^{d} k_i^2 c_{i-1}(c_i+1)$$

$$\leq 729d\log_2(n)^2\log_2(6dn)\sum_{i=1}^{d} k_i^2 c_{i-1}(c_i+1)$$

$$\leq 729d\log_2(6dn)^3\sum_{i=1}^{d} k_i^2 c_{i-1}(c_i+1)$$

For binary classifiers, generalization error can be in terms of Rademacher complexity (Mohri et al., 2012) which in turn can be bounded by $72\sqrt{\mathrm{VC}/m}$ (Kontorovich, 2016). Therefore, we can get the following[9] generalization bound:

$$L \leq \hat{L} + 144\sqrt{\frac{VC(\mathcal{F})}{m}} + \sqrt{\frac{\log(1/\delta)}{m}} \tag{17}$$

For multi-class classification, the generalization error can be similarly bounded by Graph dimension which is an extension of VC-dimension. A simple approach get a bound on Graph dimension is to consider all pairs of classes as binary classification problem which bounds the graph dimension by $\kappa^2 VC(\mathcal{F})$. There, putting everything together, we get the following generalization bound:

$$L \leq \hat{L} + 4000\kappa\sqrt{\frac{d\log_2(6dn)^3\sum_{i=1}^{d} k_i^2 c_{i-1}(c_i+1)}{m}} + \sqrt{\frac{\log(1/\delta)}{m}} \tag{18}$$

Inspired by Theorem 2, we define the following $VC$-based measure for generalization:

$$\mu_{VC}(f_{\mathbf{w}}) = \left(4000\kappa\sqrt{d\log_2(6dn)^3\sum_{i=1}^{d} k_i^2 c_{i-1}(c_i+1)} + \sqrt{\log(1/\delta)}\right)^2 \tag{19}$$

Since some of the dependencies in the above measure are probably proof artifacts, we also define another measure that is nothing but the number of parameters of the model:

$$\mu_{\text{param}} = \sum_{i=1}^{d} k_i^2 c_{i-1}(c_i+1) \tag{20}$$

### D.1.1 MEASURES ON THE OUTPUT OF THE NETWORK

While measures that can be calculated only based on the output of the network cannot reveal complexity of the network, they can still be very informative for predicting generalization. Therefore, we define a few measures that can be calculated solely based on the output of the network.

We start by looking at the cross-entropy over the output. Even though we used a cross-entropy based stopping criterion, the cross-entropy of the final models is not exactly the same as the stopping criterion and it could be informative. Hence we define the following measure:

$$\mu_{\text{cross-entropy}} = \frac{1}{m}\sum_{i=1}^{m} \ell(f_{\mathbf{w}}(\mathbf{X}_i), y_i) \tag{21}$$

where $\ell$ is the cross-entropy loss.

---

[9]The generalization gap is bounded by two times Rademacher Complexity, hence the constant 144.

Another useful and intuitive notion that appears in generalization bounds is margin. In all measures that involve margin $\gamma$, we set the margin $\gamma$ to be the 10-th percentile of the margin values on the training set and therefore ensuring $\hat{L}_\gamma \leq 0.1$. Even though margin alone is not a sensible generalization measure and can be artificially increased by scaling up the magnitude of the weights, it could still reveal information about training dynamics and therefore be informative. We report the following measure based on the margin:

$$\mu_{1/\text{margin}}(f_{\mathbf{w}}) = \frac{1}{\gamma^2} \tag{22}$$

Finally, entropy of the output is another interesting measure and it has been shown that regularizing it can improve generalization in deep learning (Pereyra et al., 2017). With a fixed cross-entropy, increasing the entropy corresponds to distribute the uncertainty of the predictions equally among the wrong labels which is connected to label smoothing and increasing the margin. We define the following measure which is the negative entropy of the output of the network:

$$\mu_{\text{neg-entropy}}(f_{\mathbf{w}}) = \frac{1}{m} \sum_{i=1}^{m} \sum_{j=1}^{\kappa} p_i[j] \log(p_i[j]) \tag{23}$$

where $p_i[j]$ is the predicted probability of the class $j$ for the input data $\mathbf{X}_i$.

### D.2 (NORM & MARGIN)-BASED MEASURES

Several generalization bounds have been proved for neural networks using margin and norm notions. In this section, we go over several such measures. For fully connected networks, Bartlett and Mendelson (2002) have shown a bound based on product of $\ell_{1,\infty}$ norm of the layer weights times a $2^d$ factor where $\ell_{1,\infty}$ is the maximum over hidden units of the $\ell_2$ norm of the incoming weights to the hidden unit. Neyshabur et al. (2015b) proved a bound based on product of Frobenius norms of the layer weights times a $2^d$ factor and Golowich et al. (2017) was able to improve the factor to $\sqrt{d}$. Bartlett et al. (2017) proved a bound based on product of spectral norm of the layer weights times sum over layers of ratio of Frobenius norm to spectral norm of the layer weights and Neyshabur et al. (2018a) showed a similar bound can be achieved in a simpler way using PAC-bayesian framework.

**Spectral Norm** Unfortunately, none of the above founds are directly applicable to convolutional networks. Pitas et al. (2017) built on Neyshabur et al. (2018a) and extended the bound on the spectral norm to convolutional networks. The bound is very similar to the one for fully connected networks by Bartlett et al. (2017). We next restate their generalization bound for convolutional networks including the constants.

**Theorem 3 (Pitas et al. (2017))** *Let $B$ an upper bound on the $\ell_2$ norm of any point in the input domain. For any $B, \gamma, \delta > 0$, the following bound holds with probability $1 - \delta$ over the training set:*

$$L \leq \hat{L}_\gamma + \sqrt{\frac{\left(84B \sum_{i=1}^{d} k_i \sqrt{c_i} + \sqrt{\ln(4n^2 d)}\right)^2 \prod_{i=1}^{d} \|\mathbf{W}_i\|_2^2 \sum_{j=1}^{d} \frac{\|\mathbf{W}_j - \mathbf{W}_j^0\|_F^2}{\|\mathbf{W}_j\|_2^2} + \ln(\frac{m}{\delta})}{\gamma^2 m}} \tag{24}$$

Inspired by the above theorem, we define the following spectral measure:

$$\mu_{\text{spec,init}}(f_{\mathbf{w}}) = \frac{\left(84B \sum_{i=1}^{d} k_i \sqrt{c_i} + \sqrt{\ln(4n^2 d)}\right)^2 \prod_{i=1}^{d} \|\mathbf{W}_i\|_2^2 \sum_{j=1}^{d} \frac{\|\mathbf{W}_j - \mathbf{W}_j^0\|_F^2}{\|\mathbf{W}_j\|_2^2} + \ln(\frac{m}{\delta})}{\gamma^2} \tag{25}$$

The generalization bound in Theorem 3 depends on reference tensors $\mathbf{W}_i^0$. We chose the initial tensor as the reference in the above measure but another reasonable choice is the origin which gives the following measures:

$$\mu_{\text{spec-orig}}(f_{\mathbf{w}}) = \frac{\left(84B \sum_{i=1}^{d} k_i \sqrt{c_i} + \sqrt{\ln(4n^2 d)}\right)^2 \prod_{i=1}^{d} \|\mathbf{W}_i\|_2^2 \sum_{j=1}^{d} \frac{\|\mathbf{W}_j\|_F^2}{\|\mathbf{W}_j\|_2^2} + \ln(\frac{m}{\delta})}{\gamma^2} \tag{26}$$

Since some of the terms in the generalization bounds might be proof artifacts, we also measure the main terms in the generalization bound:

$$\mu_{\text{spec-init-main}}(f_{\mathbf{w}}) = \frac{\prod_{i=1}^{d} \|\mathbf{W}_i\|_2^2 \sum_{j=1}^{d} \frac{\|\mathbf{W}_j - \mathbf{W}_j^0\|_F^2}{\|\mathbf{W}_j\|_2^2}}{\gamma^2} \tag{27}$$

$$\mu_{\text{spec-orig-main}}(f_{\mathbf{w}}) = \frac{\prod_{i=1}^{d} \|\mathbf{W}_i\|_2^2 \sum_{j=1}^{d} \frac{\|\mathbf{W}_j\|_F^2}{\|\mathbf{W}_j\|_2^2}}{\gamma^2} \tag{28}$$

We further look at the main two terms in the bound separately to be able to differentiate their contributions.

$$\mu_{\text{spec-init-main}}(f_{\mathbf{w}}) = \frac{\prod_{i=1}^{d} \|\mathbf{W}_i\|_2^2 \sum_{j=1}^{d} \frac{\|\mathbf{W}_j - \mathbf{W}_j^0\|_F^2}{\|\mathbf{W}_j\|_2^2}}{\gamma^2} \tag{29}$$

$$\mu_{\text{spec-orig-main}}(f_{\mathbf{w}}) = \frac{\prod_{i=1}^{d} \|\mathbf{W}_i\|_2^2 \sum_{j=1}^{d} \frac{\|\mathbf{W}_j\|_F^2}{\|\mathbf{W}_j\|_2^2}}{\gamma^2} \tag{30}$$

$$\mu_{\text{prod-of-spec/margin}}(f_{\mathbf{w}}) = \frac{\prod_{i=1}^{d} \|\mathbf{W}_i\|_2^2}{\gamma^2} \tag{31}$$

$$\mu_{\text{prod-of-spec}}(f_{\mathbf{w}}) = \prod_{i=1}^{d} \|\mathbf{W}_i\|_2^2 \tag{32}$$

$$\mu_{\text{fro/spec}}(f_{\mathbf{w}}) = \sum_{i=1}^{d} \frac{\|\mathbf{W}_i\|_F^2}{\|\mathbf{W}_i\|_2^2} \tag{33}$$

Finally, since product of spectral norms almost certainly increases with depth, we look at the following measure which is equal to the sum over squared spectral norms after rebalancing the layers to have the same spectral norms:

$$\mu_{\text{sum-of-spec/margin}}(f_{\mathbf{w}}) = d \left( \frac{\prod_{i=1}^{d} \|\mathbf{W}_i\|_2^2}{\gamma^2} \right)^{1/d} \tag{34}$$

$$\mu_{\text{sum-of-spec}}(f_{\mathbf{w}}) = d \left( \|\mathbf{W}_i\|_2^2 \right)^{1/d} \tag{35}$$

**Frobenius Norm**   The generalization bound given in Neyshabur et al. (2015b) is not directly applicable to convolutional networks. However, Since for each layer $i$, we have $\|\mathbf{W}_i\|_2 \leq k_i^2 \|\mathbf{W}_i\|_F$ and therefore by Theorem 3, we can get an upper bound on the test error based on product of Frobenius norms. Therefore, we define the following measure based on the product of Frobenius norms:

$$\mu_{\text{prod-of-fro/margin}}(f_{\mathbf{w}}) = \frac{\prod_{i=1}^{d} \|\mathbf{W}_i\|_F^2}{\gamma^2} \tag{36}$$

$$\mu_{\text{prod-of-fro}}(f_{\mathbf{w}}) = \prod_{i=1}^{d} \|\mathbf{W}_i\|_F^2 \tag{37}$$

We also look at the following measure with correspond to sum of squared Frobenius norms of the layers after rebalancing them to have the same norm:

$$\mu_{\text{sum-of-fro/margin}}(f_{\mathbf{w}}) = d \left( \frac{\prod_{i=1}^{d} \|\mathbf{W}_i\|_F^2}{\gamma^2} \right)^{1/d} \tag{38}$$

$$\mu_{\text{sum-of-fro}}(f_{\mathbf{w}}) = d \left( \prod_{i=1}^{d} \|\mathbf{W}_i\|_F^2 \right)^{1/d} \tag{39}$$

Finally, given recent evidence on the importance of distance to initialization (Dziugaite and Roy, 2017; Nagarajan and Kolter, 2019b; Neyshabur et al., 2018b), we calculate the following measures:

$$\mu_{\text{frobenius-distance}}(f_{\mathbf{w}}) = \sum_{i=1}^{d} \left\| \mathbf{W}_i - \mathbf{W}_i^0 \right\|_F^2 \tag{40}$$

$$\mu_{\text{dist-spec-init}}(f_{\mathbf{w}}) = \sum_{i=1}^{d} \left\| \mathbf{W}_i - \mathbf{W}_i^0 \right\|_2^2 \tag{41}$$

In case when the reference matrix $\mathbf{W}_i^0 = 0$ for all weights, Eq (40) the Frobenius norm of the parameters which also correspond to distance from the origin:

$$\mu_{\text{param-norm}}(f_{\mathbf{w}}) = \sum_{i=1}^{d} \left\| \mathbf{W}_i \right\|_F^2 \tag{42}$$

**Path-norm**   Path-norm was introduced in Neyshabur et al. (2015b) as an scale invariant complexity measure for generalization and is shown to be a useful geometry for optimization Neyshabur et al. (2015a). To calculate path-norm, we square the parameters of the network, do a forward pass on an all-ones input and then take square root of sum of the network outputs. We define the following measures based on the path-norm:

$$\mu_{\text{path-norm/margin}}(f_{\mathbf{w}}) = \frac{\sum_i f_{\mathbf{w}^2}(\mathbf{1})[\mathbf{i}]}{\gamma^2} \tag{43}$$

$$\mu_{\text{path-norm}}(f_{\mathbf{w}}) = \sum_i f_{\mathbf{w}^2}(\mathbf{1}) \tag{44}$$

where $\mathbf{w}^2 = \mathbf{w} \circ \mathbf{w}$ is the element-wise square operation on the parameters.

**Fisher-Rao Norm**   Fisher-Rao metric was introduced in Liang et al. (2017) as a complexity measure for neural networks. Liang et al. (2017) showed that Fisher-Rao norm is a lower bound on the path-norm and it correlates in some cases. We define a measure based on the Fisher-Rao matric of the network:

$$\mu_{\text{Fisher-Rao}}(f_{\mathbf{w}}) = \frac{(d+1)^2}{m} \sum_{i=1}^{m} \langle \mathbf{w}, \nabla_{\mathbf{w}} \ell(f_{\mathbf{w}}(\mathbf{X}_i)), y_i \rangle^2 \tag{45}$$

where $\ell$ is the cross-entropy loss.

### D.3   Flatness-based Measures

PAC-Bayesian framework (McAllester, 1999) allows us to study flatness of a solution and connect it to generalization. Given a prior $P$ is is chosen before observing the training set and a posterior $Q$ which is a distribution on the solutions of the learning algorithm (and hence depends on the training set), we can bound the expected generalization error of solutions generated from $Q$ with high probability based on the KL divergence of $P$ and $Q$. The next theorem states a simplified version of PAC-Bayesian bounds.

**Theorem 4** *For any $\delta > 0$, distribution $D$, prior $P$, with probability $1 - \delta$ over the training set, for any posterior $Q$ the following bound holds:*

$$\mathbb{E}_{\mathbf{v} \sim Q}[L(f_{\mathbf{v}})] \leq \mathbb{E}_{\mathbf{w} \sim Q}\left[\hat{L}(f_{\mathbf{v}})\right] + \sqrt{\frac{\text{KL}(Q\|P) + \log\left(\frac{m}{\delta}\right)}{2(m-1)}} \tag{46}$$

If $P$ and $Q$ are Gaussian distributions with $P = \mathcal{N}(\mu_P, \Sigma_P)$ amd $Q = \mathcal{N}(\mu_Q, \Sigma_Q)$, then the KL-term can be written as follows:

$$\text{KL}(\mathcal{N}(\mu_Q, \Sigma_Q)\|\mathcal{N}(\mu_P, \Sigma_P)) = \frac{1}{2}\left[\text{tr}\left(\Sigma_P^{-1}\Sigma_Q\right) + (\mu_Q - \mu_P)^\top \Sigma_P^{-1}(\mu_Q - \mu_P) - k + \ln(\frac{\det \Sigma_P}{\det \Sigma_Q})\right].$$

Setting $Q = \mathcal{N}(\mathbf{w}, \sigma^2 I)$ and $P = \mathcal{N}(\mathbf{w}^0, \sigma^2 I)$ similar to Neyshabur et al. (2017), the KL term will be simply $\frac{\|\mathbf{w}-\mathbf{w}^0\|_2^2}{2\sigma^2}$. However, since $\sigma$ belongs to prior, if we search to find a value for $\sigma$, we need to adjust the bound to reflect that. Since we search over less than 20000 predefined values of $\sigma$ in our experiments, we can use the union bound which changes the logarithmic term to $\log(20000m/\delta)$ and we get the following bound:

$$\mathbb{E}_{\mathbf{u}\sim\mathcal{N}(u,\sigma^2 I)}\left[L(f_{\mathbf{w}+\mathbf{u}})\right] \leq \mathbb{E}_{\mathbf{u}\sim\mathcal{N}(u,\sigma^2 I)}\left[\hat{L}(f_{\mathbf{w}+\mathbf{u}})\right] + \sqrt{\frac{\frac{\|\mathbf{w}-\mathbf{w}^0\|_2^2}{4\sigma^2} + \log(\frac{m}{\sigma}) + 10}{m-1}} \tag{47}$$

Based on the above bound, we define the following measures using the origin and initialization as reference tensors:

$$\mu_{\text{pac-bayes-init}}(f_{\mathbf{w}}) = \frac{\|\mathbf{w}-\mathbf{w}^0\|_2^2}{4\sigma^2} + \log(\frac{m}{\sigma}) + 10 \tag{48}$$

$$\mu_{\text{pac-bayes-orig}}(f_{\mathbf{w}}) = \frac{\|\mathbf{w}\|_2^2}{4\sigma^2} + \log(\frac{m}{\delta}) + 10 \tag{49}$$

where $\sigma$ is chosen to be the largest number such that $\mathbb{E}_{\mathbf{u}\sim\mathcal{N}(u,\sigma^2 I)}\left[\hat{L}(f_{\mathbf{w}+\mathbf{u}})\right] \leq 0.1$.

The above framework captures flatness in the expected sense since we add Gaussian perturbations to the parameters. Another notion of flatness is the worst-case flatness where we search for the direction that changes the loss the most. This is motivated by (Keskar et al., 2016) where they observe that this notion would correlate to generalization in the case of different batch sizes. We can use PAC-Bayesian framework to give generalization bounds for worst-case perturbations as well. The magnitude of a Gaussian variable with with variance $\sigma^2$ is at most $\sigma\sqrt{2\log(2/\delta)}$ with probability $1-\delta/2$. Applying a union bound on all parameters, we get that with probability $1-\delta/2$ the magnitude of the Gaussian noise is at most $\alpha = \sigma\sqrt{2\log(2\omega/\delta)}$ where $\omega$ is the number of parameters of the model. Therefore, we can get the following generalization bound:

$$\mathbb{E}_{\mathbf{u}\sim\mathcal{N}(u,\sigma^2 I)}\left[L(f_{\mathbf{w}+\mathbf{u}})\right] \leq \max_{|u_i|\leq\alpha}\hat{L}(f_{\mathbf{w}+\mathbf{u}}) + \sqrt{\frac{\frac{\|\mathbf{w}-\mathbf{w}^0\|_2^2\log(2\omega/\delta)}{2\alpha^2} + \log(\frac{2m}{\delta}) + 10}{m-1}} \tag{50}$$

Inspired by the above bound, we define the following measures:

$$\mu_{\text{sharpness-init}}(f_{\mathbf{w}}) = \frac{\|\mathbf{w}-\mathbf{w}^0\|_2^2\log(2\omega)}{4\alpha^2} + \log(\frac{m}{\sigma}) + 10 \tag{51}$$

$$\mu_{\text{sharpness-orig}}(f_{\mathbf{w}}) = \frac{\|\mathbf{w}\|_2^2\log(2\omega)}{4\alpha^2} + \log(\frac{m}{\delta}) + 10 \tag{52}$$

where $\alpha$ is chosen to be the largest number such that $\max_{|u_i|\leq\alpha}\hat{L}(f_{\mathbf{w}+\mathbf{u}}) \leq 0.1$.

To understand the importance of the flatness parameters $\sigma$ and $\alpha$, we also define the following measures:

$$\mu_{\text{pac-bayes-flatness}}(f_{\mathbf{w}}) = \frac{1}{\sigma^2} \tag{53}$$

$$\mu_{\text{sharpness-flatness}}(f_{\mathbf{w}}) = \frac{1}{\alpha^2} \tag{54}$$

where $\alpha$ and $\sigma$ are computed as explained above.

**Magnitude-aware Perturbation Bounds** The magnitude of perturbation in (Keskar et al., 2016) was chosen so that for each parameter the ratio of magnitude of perturbation to the magnitude of the parameter is bounded by a constant $\alpha'$[10]. Following a similar approach, we can choose the posterior for parameter $i$ in PAC-Bayesian framework to be $\mathcal{N}(w_i, \sigma'^2|w_i|^2 + \epsilon^2)$. Now, substituting this

---

[10]They actually used a slightly different version which is a combination of the two perturbation bounds we calculated here. Here, for more clarity, we decomposed it into two separate perturbation bounds.

in the Equation equation D.3 and solving for the prior $\mathcal{N}(\mathbf{w}^0, \sigma_P^2)$ that minimizes the KL term by setting the gradient with respect to $\sigma_2^P$ to zero, KL can be written as follows:

$$2\text{KL}(Q\|P) = \omega \log\left(\frac{\sigma'^2+1}{\omega}\left\|\mathbf{w}-\mathbf{w}^0\right\|_2^2 + \epsilon^2\right) - \sum_{i=1}^{\omega}\log\left(\sigma'^2|w_i - w_i^0|^2 + \epsilon^2\right)$$

$$= \sum_{i=1}^{\omega}\log\left(\frac{\epsilon^2 + (\sigma'^2+1)\left\|\mathbf{w}-\mathbf{w}^0\right\|_2^2/\omega}{\epsilon^2 + \sigma'^2|w_i - w_i^0|^2}\right)$$

Therefore, the generalization bound can be written as follows

$$\mathbb{E}_{\mathbf{u}}\left[L(f_{\mathbf{w}+\mathbf{u}})\right] \leq \mathbb{E}_{\mathbf{u}}\left[\hat{L}(f_{\mathbf{w}+\mathbf{u}})\right] + \sqrt{\frac{\frac{1}{4}\sum_{i=1}^{\omega}\log\left(\frac{\epsilon^2 + (\sigma'^2+1)\|\mathbf{w}-\mathbf{w}^0\|_2^2/\omega}{\epsilon^2 + \sigma'^2|w_i - w_i^0|^2}\right) + \log(\frac{m}{\delta}) + 10}{m-1}} \quad (55)$$

where $u_i \sim \mathcal{N}(0, \sigma'^2|w_i| + \epsilon^2)$, $\epsilon = 1e-3$ and $\sigma'$ is chosen to be the largest number such that $\mathbb{E}_{\mathbf{u}}\left[\hat{L}(f_{\mathbf{w}+\mathbf{u}})\right] \leq 0.1$. We define the following measures based on the generalization bound:

$$\mu_{\text{pac-bayes-mag-init}}(f_{\mathbf{w}}) = \frac{1}{4}\sum_{i=1}^{\omega}\log\left(\frac{\epsilon^2 + (\sigma'^2+1)\left\|\mathbf{w}-\mathbf{w}^0\right\|_2^2/\omega}{\epsilon^2 + \sigma'^2|w_i - w_i^0|^2}\right) + \log(\frac{m}{\delta}) + 10 \quad (56)$$

$$\mu_{\text{pac-bayes-mag-orig}}(f_{\mathbf{w}}) = \frac{1}{4}\sum_{i=1}^{\omega}\log\left(\frac{\epsilon^2 + (\sigma'^2+1)\left\|\mathbf{w}\right\|_2^2/\omega}{\epsilon^2 + \sigma'^2|w_i - w_i^0|^2}\right) + \log(\frac{m}{\delta}) + 10 \quad (57)$$

We also follow similar arguments are before to get a similar bound on the worst-case sharpness:

$$\mathbb{E}_{\mathbf{u}}\left[L(f_{\mathbf{w}+\mathbf{u}})\right] \leq \max_{|u_i|\leq\alpha'|w_i|+\epsilon}\hat{L}(f_{\mathbf{w}+\mathbf{u}}) + \sqrt{\frac{\frac{1}{4}\sum_{i=1}^{\omega}\log\left(\frac{\epsilon^2 + (\alpha'^2+4\log(2\omega/\delta))\|\mathbf{w}-\mathbf{w}^0\|_2^2/\omega}{\epsilon^2 + \alpha'^2|w_i - w_i^0|^2}\right) + \log(\frac{m}{\delta}) + 10}{m-1}}$$
$$(58)$$

We look at the following measures based on the above bound:

$$\mu_{\text{pac-sharpness-mag-init}}(f_{\mathbf{w}}) = \frac{1}{4}\sum_{i=1}^{\omega}\log\left(\frac{\epsilon^2 + (\alpha'^2 + 4\log(2\omega/\delta))\left\|\mathbf{w}-\mathbf{w}^0\right\|_2^2/\omega}{\epsilon^2 + \alpha'^2|w_i - w_i^0|^2}\right) + \log(\frac{m}{\delta}) + 10$$
$$(59)$$

$$\mu_{\text{pac-sharpness-mag-orig}}(f_{\mathbf{w}}) = \frac{1}{4}\sum_{i=1}^{\omega}\log\left(\frac{\epsilon^2 + (\alpha'^2 + 4\log(2\omega/\delta))\left\|\mathbf{w}\right\|_2^2/\omega}{\epsilon^2 + \alpha'^2|w_i - w_i^0|^2}\right) + \log(\frac{m}{\delta}) + 10$$
$$(60)$$

Finally, we look at measures that are only based the sharpness values computed above:

$$\mu_{\text{pac-bayes-mag-flat}}(f_{\mathbf{w}}) = \frac{1}{\sigma'^2} \quad (61)$$

$$\mu_{\text{sharpness-mag-flat}}(f_{\mathbf{w}}) = \frac{1}{\alpha'^2} \quad (62)$$

where $\alpha$ and $\sigma$ are computed as explained above.

## D.4 OPTIMIZATION-BASED MEASURES

There are mixed results about how the optimization speed is relevant to generalization. On one hand we know that adding Batch Normalization or using shortcuts in residual architectures help both optimization and generalization and Hardt et al. (2015) suggests that faster optimization results in better generalization. On the other hand, there are empirical results showing that adaptive optimization methods that are faster, usually generalize worse (Wilson et al., 2017b). Here, we put these hypothesis into test by looking at the number of steps to achieve cross-entropy 0.1 and the number of steps needed to go from cross-entropy 0.1 to 0.01:

$$\mu_{\text{\#steps-0.1-loss}}(f_{\mathbf{w}}) = \text{\#steps from initialization to 0.1 cross-entropy} \tag{63}$$

$$\mu_{\text{\#steps-0.1-0.01-loss}}(f_{\mathbf{w}}) = \text{\#steps from 0.1 to 0.01 cross-entropy} \tag{64}$$

The above measures tell us if the speed of optimization at early or late stages can be informative about generalization. We also define measures that look at the SGD gradient noise after the first epoch and at the end of training at cross-entropy 0.01 to test the gradient noise can be predictive of generalization:

$$\mu_{\text{grad-noise-epoch1}}(f_{\mathbf{w}}) = \text{Var}_{(\mathbf{X},y)\ S} \left( \nabla_{\mathbf{w}} \ell(f_{\mathbf{w}^1}(\mathbf{X}), y) \right) \tag{65}$$

$$\mu_{\text{grad-noise-final}}(f_{\mathbf{w}}) = \text{Var}_{(\mathbf{X},y)\ S} \left( \nabla_{\mathbf{w}} \ell(f_{\mathbf{w}}(\mathbf{X}), y) \right) \tag{66}$$

where $\mathbf{w}^1$ is the weight vector after the first epoch.

## E  ALGORITHMS

We first lay out some common notations used in the pseudocode:

1. $f$: the architecture that takes parameter $\theta$ and input $x$ and map to $f(x; \theta)$ which is the predicted label of $x$

2. $\theta$: parameters

3. $M$: Some kind of iteration; $M_1$: binary search depth; $M_2$: Monte Carlo Estimation steps; $M_3$: Iteration for estimating the loss

4. $\mathscr{D} = \{(x_i, y_i)\}_{i=0}^n$ the dataset the model is trained on; $\mathcal{B}$ as a uniformly sampled minibatch from the dataset.

Both search algorithm relies on the assumption that the loss increases monotonically with the perturbation magnitude $\sigma$ around the final weight. This assumption is quite mild and in reality holds across almost all the models in this study.

---

**Algorithm 1** EstimateAccuracy

---

1: **Inputs:** model $f$, parameter $\theta$, dataset $\mathscr{D}$, estimate iteration $M$
2: **Initialize** Accuracy = 0
3: **for** episode $i = 1$ **to** $M$ **do**
4:     $\mathcal{B} \sim \text{sample}(\mathscr{D})$
5:     Accuracy += $\frac{1}{|\mathcal{B}|} \sum_i \delta(y_i = f(\mathcal{B}_i; \theta))$
6: **end for**
7: **return** Accuracy/M

---

Note that for finding the sharpness $\sigma$, we use the cross-entropy as the differentiable surrogate object instead of the 1-0 loss which is in general not differentiable. Using gradient ascent brings another additional challenge that is for a converged model, the local gradient signal is usually weak, making gradient ascent extremely inefficient. To speed up thie process, we add a uniform noise with range being $[-\sigma_{new}/N_{\mathbf{w}}, \sigma_{new}/N_{\mathbf{w}}]$ to lift the weight off the flat minima where $N_{\mathbf{w}}$ is the number of parameters. This empirical greatly accelerates the search.

Further, for magnitude aware version of the bounds, the overall algorithm stays the same with the exception that now covariance matrices at line 7 of Algorithm 2 become as diagonal matrix containing $w_i^2$ on the diagonal; similarly, for line 12 of Algorithm 3, the weight clipping of each $w_i$ is conditioned on $\sigma_{new}|w_i|$, i.e. clipped to $[-\sigma_{new}|w_i|, \sigma_{new}|w_i|]$. Here $w_i$ denotes the $i^{th}$ parameter of flattened $\mathbf{w}$.

---

**Algorithm 2** Find $\sigma$ for PAC-Bayesian Bound

---

1: **Inputs:** $f$, $\theta_0$, model accuracy $\ell$, target accuracy deviation $d$, Upper bound $\sigma_{\max}$, Lower bound $\sigma_{\min}$, $M_1$, $M_2$, $M_3$
2: **Initialize**
3: **for** episode $i = 1$ **to** $M_1$ **do**
4:     $\sigma_{new} = (\sigma_{\max} + \sigma_{\min})/2$
5:     $\hat{\ell} = 0$
6:     **for** step $j = 0$ **to** $M_2$ **do**
7:        $\theta \leftarrow \theta_0 + \mathcal{N}(0, \sigma_{new}^2 I)$
8:        $\hat{\ell} = \hat{\ell} + \text{EstimateAccuracy}(f, \theta_{new}, \mathscr{D}, M_3)$
9:     **end for**
10:    $\hat{\ell} = \hat{\ell}/M_2$
11:    $\hat{d} = |\ell - \hat{\ell}|$
12:    **if** $\hat{d} < \epsilon_d$ **or** $\sigma_{\max} - \sigma_{\min} < \epsilon_\sigma$ **then**
13:       **return** $\sigma_{new}$
14:    **end if**
15:    **if** $\hat{d} > d$ **then**
16:       $\sigma_{\max} = \sigma_{new}$
17:    **else**
18:       $\sigma_{\min} = \sigma_{new}$
19:    **end if**
20: **end for**

---

**Algorithm 3** Find $\sigma$ for Sharpness Bound

---

1: **Inputs:** $f$, $\theta_0$, loss function $\mathcal{L}$, model accuracy $\ell$, target accuracy deviation $d$, Upper bound $\sigma_{\max}$, Lower bound $\sigma_{\min}$, $M_1$, $M_2$, $M_3$, gradient steps $M_4$
2: **Initialize**
3: **for** episode $i = 1$ **to** $M_1$ **do**
4:     $\sigma_{new} = (\sigma_{\max} + \sigma_{\min})/2$
5:     $\hat{\ell} = \infty$
6:     **for** step $j = 0$ **to** $M_2$ **do**
7:        $\theta = \theta_0 + \mathcal{U}(\sigma_{new}/2)$
8:        **for** step $k = 0$ **to** $M_4$ **do**
9:           $\mathcal{B} \sim \text{sample}(\mathscr{D})$
10:          $\theta = \theta + \eta\nabla_\theta \ell(f, \mathcal{B}, \theta)$
11:          **if** $||\theta|| > \sigma_{new}$ **then**
12:             $\theta = \sigma_{new} \cdot \frac{\theta}{||\theta||}$
13:          **end if**
14:        **end for**
15:        $\hat{\ell} = \min(\hat{\ell}, \text{EstimateAccuracy}(f, \theta_{new}, \mathscr{D}, M_3))$
16:     **end for**
17:    $\hat{d} = |\ell - \hat{\ell}|$
18:    **if** $\hat{d} < \epsilon_d$ **or** $\sigma_{\max} - \sigma_{\min} < \epsilon_\sigma$ **then**
19:       **return** $\sigma_{new}$
20:    **end if**
21:    **if** $\hat{d} > d$ **then**
22:       $\sigma_{\max} = \sigma_{new}$
23:    **else**
24:       $\sigma_{\min} = \sigma_{new}$
25:    **end if**
26: **end for**

---

