# OpenReview forum: "Fantastic Generalization Measures and Where to Find Them"
_ICLR.cc/2020/Conference — Accept (Poster)_

### Official Review · AnonReviewer1 · 2019-10-21
**Official Blind Review #1**

**Rating:** 8

**Review:**

This work all starts with the observation, made by many, that deeper (>=2 layers) networks generalise differently to the simple models we used to use, and that the good old bias-variance trade-off doesn't really apply.

This paper is a natural successor to a long line of papers, most recently Zhang at el (best paper at ICLR 2017, though some like Tom Dietterich weren't impressed), Neyshabur et al (NIPS 2017) and Jiang et al (ICLR 2019).  Whereas Jiang etal and Neyshabur etal had a few metrics, a few data sets and a few models, this paper instead has loads and loads of metrics and just one data set and one model class.  You have, however, been clearly strongly influenced by the prior work and attempted to fix what you saw as their empirical shortcomings.

I agree with the choice of stopping according to cross entropy.  One thing not done in Section 5 is a discussion of
variation:  choice of initialisation and the impact of stochasticity in the optimisation.

Appendix C lists your copious metrics.  I am not deeply embedded in the theory, though I understand the major concepts. I was impressed with the coverage and variations done.  I expect Appendix C may raise copius matching arguments amongst the theory community, with some valid and some invalid complaints.  For me, I wonder why you make P & Q have the same variance (see before equation 43 in App. C).  You credit Neyshabur, but I'm left wondering.

So, the short version of my review is that I believe this paper is imcomplete, in that not enough different data sets were investigated.  With this, I fear you could well be uncovering perculiarities pertinant to CIFAR-10 and the ease of getting near perfect training error on it.  However, I am impressed with what you have done, and think you made a great starting point.  So I say, publish and let the real battle begin.  Let the theoreticians argue (about metrics) and the practicioners implement (with more data), and let's see what happens.  You have laid a great ground work for folks, bulding on the earlier work.  Perhaps Google can be encouraged to support this in that Jiang's paper is a precursor.

Interesting observations in Appendix A.1, as are some of the discussions in Sections 7, 8, 9.

Given the best metric seems to be sharpness, which exceeds PAC-Bayesian, shouldn't you relate this to grad noise (metrics 61 and 62) which measure the flatness.  Are these related?  Note, also, a Bayesian always wants broad peaks for their parameter surfaces.

MINOR ISSUES: (1) some repeated words: "an an", "the the" (2) some strange grammar "each canonical ordering that only be predictive for its own category", "VC-dimension bounds or and parameter counting", "possible to find to calculate", "in the Equation equation C.3 (3)" Generalisation gap is poorly introduced.  Its officially defined in footnote 4 but used way earlier!  Put in the main text:  its an important clarification. (4) Table 1 legend should mention the "numerical results" are for equations 2, 4 and 5.  (5)  Somewhere you need to mention that numbers in red in tables refer to equations in appendix C.


**Experience Assessment:**

I have published one or two papers in this area.

**Review Assessment: Checking Correctness Of Derivations And Theory:**

I assessed the sensibility of the derivations and theory.

**Review Assessment: Checking Correctness Of Experiments:**

I carefully checked the experiments.

**Review Assessment: Thoroughness In Paper Reading:**

I read the paper at least twice and used my best judgement in assessing the paper.

---

> ### Author Response · Authors · 2019-11-13
> **Author Response to Review #1**
>
> Thank you for your encouraging review! We next address your questions:
>
>     1) The stochasticity due to initialization and optimization: We have added this discussion to the revision. We ran all experiments 5 times and calculated the standard deviation (table 8&9). The resulting standard deviation shows that our result is statistically significant. Please see the paper for more discussion.
>
>     2) All results are reported on CIFAR10: Have now repeated all experiments on Street View House Numbers (SVHN) dataset and added the result to the revision  (table 6). This result is consistent with that of CIFAR10 which shows that our conclusions are not limited to CIFAR10. In order to improve the strength of the paper even further, we are currently running similar experiments on STL10 and CIFAR100 which will be added to the final version.
>
>     3) Making make P & Q have the same variance equation 43 in App. C: Similar to Neyshabur et al, we mainly do this since it makes the derivation simpler and the final bound more interpretable. That is however not the optimal choice.
>
>     4) Relationship between sharpness and grad noise: You are right that these quantities seem very related. We indeed believe that this relationship is very interesting and can be a subject of future study!
>
> Finally, thank you for the minor suggestions. We have incorporated all of them in the revision.

---

### Official Review · AnonReviewer3 · 2019-10-25
**Official Blind Review #3**

**Rating:** 3

**Review:**

The paper aims at providing a better understanding of generalization for Deep Learning models. The idea is really interesting for the ML community as, despite their broad use, the astounding property of deep neural networks to generalize that well is still not well understood.
The idea is not to show new theoretical bounds for generalization gaps but stress the results of an empirical study comparing the already existing measures. The authors choose 7 common hyperparameters related to optimization and analyze the correlation between the generalization  gaps effectively observed and the ones predicted by different measures (VC dim, cross-entropy, canonical ordering …).
The writing of the paper is clear and easily understandable. Besides, I believe that the study is relevant to ICLR conference.

However, I believe the level of the paper is marginally below the threshold of acceptance and therefore would recommend to reject it.

The paper is solely empirical but I believe that the empirical section is a bit weak, or at least some important points remain unclear. If I appreciate the extent efforts made in trying to evaluate different measures of generalization gaps, I do not believe that the findings are conclusive enough.

1) First, all this empirical result are based on one-dataset (CIFAR-10) only thus limiting the impact of the study. Indeed, a given measure might very correlate with generalization gap on this specific dataset but not on others.

2) Specifically, we see that on this specific dataset, all training accuracies are already quite good (cf : Figure 1, distribution of the training losses). Consequently, authors are more correlating the chosen measures with the test error rather than with the generalization gaps. On other more complicated datasets where the training loss is higher, the VC dimension might consequently have way better results.
Similarly, in Section 6, the authors say that the results «confirm the widely known empirical observation that over-parametrization improves generalization in deep learning. » In this specific case, no reference was given to support the claim. I would agree with the claim « over-paramatrization improves test accuracy (reduces test error) » but the link between over-parametrization and generalization is less clear.

3) In Section 4, the authors say « drawing conclusion from changing one or two hyper-parameters » can be a pitfall as « the hyper-parameter could be the true cause of both change in the measure and change in the generalization ». I totally agree with the authors here. Consequently, I do not understand why the correlations were measured by only changing one hyper-parameter at a time instead of sampling randomly in Theta.

4) It is still not clear to me how the authors explain why some measures are more correlated with generalization gaps than others. Are some bounds tighter than others ? This empirical study was only applied to convolutional neural networks and consequently one may wonder that for example the VC dim bounds computed in the specific case of neural networks are too loose. However, this measure could be efficient for type of models.


I would like the authors to clear the following points :
- How do you ensure that the empirical study clearly correlated measures predictions with generalization gaps and not simply with test errors (or accuracies) ? (point 2)
- Could you please also answer Point 4 ?
- Finally, how would you explain the fact that the canonical order performs so well compare to many other measures and that it is a really tough-to-beat baseline ?



**Experience Assessment:**

I have read many papers in this area.

**Review Assessment: Checking Correctness Of Derivations And Theory:**

I assessed the sensibility of the derivations and theory.

**Review Assessment: Checking Correctness Of Experiments:**

I assessed the sensibility of the experiments.

**Review Assessment: Thoroughness In Paper Reading:**

I read the paper at least twice and used my best judgement in assessing the paper.

---

> ### Author Response · Authors · 2019-11-13
> **Author Response to Review #3**
>
> Thank you for your thoughtful feedback. We have added many more experiments and addressed all of your concerns. We hope that after looking at the revision and reading our response, you would consider increasing your score.
>
> We will next address your concerns in details below:
>
>     1) We have significantly improved the strength of our empirical results in the revision. More specifically, we repeated all experiments for Street View House Numbers (SVHN) dataset  (table 6). Furthermore, we ran all experiments on CIFAR-10 5 times and calculated the standard deviation over different runs. Both these results are indeed consistent with the observations we made on CIFAR-10. Note that this means training over 10K networks! Please see the revision for details. Even though we believe current experimental results are more than enough to verify the conclusions in the paper, we are currently running additional experiments on CIFAR100 and STL10 datasets to make the experiments even more comprehensive. Given the scale of the experiments, these results would be available for the final version.
>
>     2) When studying generalization, we follow the common practice in the field which mainly focuses on over-parameterized regime where the model capacity is high enough to fit the training data. In this regime, the test error and generalization gap become the same. We believe this regime is more interesting as in the under-parametrized regime the test error can be usually improved by using better optimization algorithms and hence generalization is less mysterious. We agree with you that observations in over-parameterized regime might not generalize to the under-parametrized regime. However, in order to address your concern about test error vs generalization, we included the results of CIFAR-10 at cross-entropy=0.1 (table 7) in the revision where the training errors are much higher and the observations are largely consistent with cross-entropy=0.01, indicating that we are indeed capturing the generalization gap rather than test accuracy.
>
>     3) We believe there might be a confusion here. What we stated is that only changing a single hyperparameter and only looking at correlation when changing one single hyperparameter can be misleading. Instead, we do controlled experiments on seven hyperparameters, meaning that we change them one by one (and each time, we change one, the other six hyperparameters are kept fixed). The expected performance in the controlled experiments are much better indicator compared to only changing one hyperparameters or changing all of them together. You also asked why we didn’t sample all hyperparameters together. We actually do report that as well in the table column called “overall $\tau$”. Please see section 2.3 for discussions on why the controlled experiments are preferred to overall $\tau$. Finally, we added even further analysis using conditional mutual information in the appendix B that is consistent with the rank correlation results reported in the main paper.
>
>     4) Not all these measures can be converted to a bound. Furthermore, a tighter bound does not necessarily correlate better with generalization. That is mainly because all current bounds are very loose so their tightness relative to each other is not informative. What we try to study is whether these measures can predict the observed generalization rather than bounding it. We chose convolutional neural networks because they are heavily used in both research and in industry. We agree with you that these results might not generalize to other types of networks such as recurrent neural networks and we have made this clear on the conclusion of the original submission. However, we believe the protocols we establish in this work will be useful for future research on other types of networks.
>
>     5) Canonical order is not a single measure but rather a number specific to the changes made to a single hyperparameter, which means that there are 7 canonical measures (please see Section 4 for definition). Hence, each canonical measure is only predictive when we change its corresponding hyperparameter and is uncorrelated with generalization when we change any other hyperparameter. Therefore, each of these measures perform poorly overall.

---

### Official Review · AnonReviewer2 · 2019-11-02
**Official Blind Review #2**

**Rating:** 8

**Review:**

The paper addresses the well-known generalization problem for deep networks: why do these models successfully generalize even though they contain more parameters than the size of the training set? Recent works on this problem have proposed various measures of model complexity (that is, different from the number of parameters) with the claim that they (at least partially) capture the "true" capacity of deep networks in order to answer this question. The authors report the results of a large-scale, systematic study of the empirical effect of these complexity measures on the observed generalization behavior of deep networks.

The empirical study of generalization behavior is inherently tricky, and the authors make a non-trivial contribution just by designing an experiment that can convincingly address this question. The authors give careful attention to various potential obstacles related to experimental design and statistical validity.

The results of the authors' experiments are highly valuable to the research community. I recommend to accept.

*********************************************************************************

Technical comments for the authors:
-Please rewrite or elaborate on the first two paragraphs of section 6 for clarity.

-It seems to make little sense to consider norm-based complexity measures that are not margin-normalized (or at least normalized in some other way). Please include a margin-normalized version of each such measure in Table 5; certain key quantities (e.g. "frob-distance") have no corresponding margin-normalized entry.

-The quantity "sum-of-frob/margin" is certainly a quantity of interest and it should be included in table 2 and possibly merits some discussion.

-I disagree somewhat with the discussion of the spectral bound in section 7. In particular, I would not agree that the poor performance of "spec-orig" is surprising, because it contains an astronomically-sized "proof artifact" (as you call it). Namely, in this bound and in similar norm-based bounds, the term which is the product of the matrices in the network is (heuristically) unnecessary; in particular it (heuristically) ought to be able to be replaced with the network's "average-case Lipschitz constant" (this is one of the issues which Arora et al. 2018 attempted to address; unfortunately formalizing an adequate notion of "average-case Lipschitz constant" can be quite cumbersome). This superfluous product of matrix norms is of course highly correlated with model size, so it is not surprising that the quantity would have a sizable negative correlation with generalization. The quantity I would consider to be the "main term" is the quantity "fro/spec". Furthermore, the fact that this quantity (which appears in both Bartlett et al. and Arora et al.) performs somewhat worse than simply "sum-of-fro" is fairly interesting and possibly merits some discussion.


**Experience Assessment:**

I have read many papers in this area.

**Review Assessment: Checking Correctness Of Derivations And Theory:**

I assessed the sensibility of the derivations and theory.

**Review Assessment: Checking Correctness Of Experiments:**

I assessed the sensibility of the experiments.

**Review Assessment: Thoroughness In Paper Reading:**

I read the paper thoroughly.

---

> ### Author Response · Authors · 2019-11-13
> **Author Response to Review #2**
>
> Thank you for your encouraging and thoughtful comments! We added more details to section 6 and address your other concerns below:
>
>      1) Margin normalization: We agree that these measures should be normalized. However, we want to point out that there is an implicit normalization effect by the fact that we stop at a certain cross-entropy which is closely related to the margin when training error is zero or very small. Therefore, the margin values of all these networks are similar (but not the same). That being said, we have normalized all quantities that usually require normalization. If you look at the normalized versions of the measures, they do not behave significantly different than the unnormalized ones. The reason we do not Euclidean distance measure is that they usually shows up in bounds together with other norm-based terms and the right way to normalize it is not clear (again since the cross-entropy and margin values are almost the same, we don’t think this makes a big difference).
>
>     2) Including sum-of-frob/margin: Thanks for pointing this out. We agree the quantity normalized by margin is important and have included it in table 2, and added corresponding discussion in the first paragraph of page 8.
>
>     3) Negative correlation of spectral bound: We agree that it is not surprising that spectral bound would be extremely large and it is not surprising if it doesn’t correlate with generalization. However, we believe that the negative correlation is very surprising. Note that the negative correlation is constantly present in the controlled experiments when two model different only differ in learning rate, or batch size, or dropout, or optimization algorithm. Therefore, this negative correlation cannot be relevant to the model size since in all these cases, two models have the exact same architecture.
>
>     4) Sum-of-fro: We completely agree with you that the main term is fro/spec and we are surprised that it performed worse than sum-of-fro. This perhaps can be a subject of future study.

---

### Public Comment · ~Ilya_Tolstikhin1 · 2019-09-26
**Whether generalization gap is what we should be looking at**

Thank you for the paper. I enjoyed reading it. I wanted to know your opinion on the following point.

You define the generalization gap as the difference between the train and test accuracy and this gap plays the central role in the whole empirical study (i.e. in conclusions regarding which hyper-parameter values are beneficial and which are not). In short, the paper assumes that the smaller this gap, the better. I would like to argue that what is more important (and what the paper actually looks at) is the test accuracy, not the generalization gap. Notice that by design all the models in the study achieved very small training loss (high training accuracy). According to Figure 2 in appendix, majority of the models achieved training misclassification error of less than 1%. This means, generalization gap takes the form of $acc_train - acc_test$, where $acc_train$ takes values in [0.99, 1]. In that context saying "small generalization gap" is equivalent to saying "large test accuracy".

I am pointing this out because in many other contexts these two statements ("small generalization gap" / "large test accuracy") may actually have two different meanings. Indeed, it seems that in most of the interesting applications DNNs manage to achieve the perfect training accuracy and thus the argument above holds (i.e. the two statements are equivalent). But I feel, what we are really looking for are the cases when we achieve a good test accuracy. For DNNs it just happens so that in those cases the generalization gap is very small. But small generalization gap obviously does not imply that the training succeeded: think of cases where the hyper-parameters are completely off and you end up with accuracy at random level both at training and test: the generalization gap is zero, but the training failed and there is no generalization.

That being said, your experimental design (looking only at models with high training accuracy) resolves this problem and indeed makes you focus on the good test performance. Nevertheless, I feel that making this point explicit (or even switching from "generalization gap" to "test loss" in your discussions) could be very useful in helping future follow up works (which I am sure will arrive sooner or later) to focus on important questions / numbers.

Small typos:
Page 5: "this this", "to choose ensure"
Page 8: sentence starting with "We next look at...".
Page 8 "to find to calculate"

---

> ### Author Response · Authors · 2019-09-27
> **Thank you for your comment**
>
> Thank you so much for reading our paper and posting your feedback (on the second day after submission!) We are glad that you liked our paper!
>
> We agree with your comment that in many applications we are interested in the test error rather than he generalization gap. As you pointed out yourself, we chose a setting where training error is almost zero so that these two coincide. This choice was indeed intentional to avoid choosing between test error and the generalization gap. A very interesting observation that we had during our experiments was that we realized if we do not control for the training accuracy, then by simply looking at the training loss one could easily predict which model will generalize better: while this is not always true, in our case we observed that at convergence models with higher loss generalize better. That made us choose training performance as a stopping criterion and look at the generalization gap. Another potential choice was to stop at a higher training error say 0.1 in which case the test error and generalization error only differ by 0.1 for all models, instead of 0.01. One issue with that choice is that there is not enough variation in the generalization of different models at that training accuracy as the models are not close to convergence. Another issue is that since the training error improves with a fast rate at each update around 0.1, it is almost impossible to stop the optimization at that exact stopping criterion. Also, one could argue that at training error 0.1, the main bottleneck to improve the test error is to fit training data better and even though this is important, it is not the focus of this study. All these difficulties led us to believe that the current setting is the best choice. We will add more discussions around this issue in next update.
>
> Once again, thank you very much for reading the paper and catching the typos, and for the encouraging comment!

---

### Author Response · Authors · 2019-11-13
**Revision Summary**

We thank all reviewers for their valuable feedback. We have updated the paper to include new experiments and revisions suggested.
    1. We repeated all the experiments in the paper on SVHN and show that the result is consistent with what we already observed on CIFAR-10 (Table 6).
    2. We repeated all the experiments on CIFAR-10 5 times and show that the observation is robust to stochasticity of initialization and optimization (mean in Table 8 and std in Table 9).
    3. We compute the same statistics for CIFAR-10 at cross-entropy=0.1 and the performance of the majority of the measures (most notably spectral and sharpness) are consistent with cross-entropy=0.01. This shows that we are indeed capturing the generalization gap rather than test error (Table 7).
    4. We have added more analysis in the appendix B using conditional mutual information. The results are consistent with the ones reported in the main paper.
    5. Improved writing including related work, notation etc and incorporated all the suggestions made by reviewer 1&2.
Merged related work and subsection into subsections of section 1

---

### Decision · Program_Chairs · 2019-12-19

**Decision:**

Accept (Poster)

**Comment:**

This paper provides a valuable survey, summary, and empirical comparison of many generalization quantities from throughout the literature. It is comprehensive, thorough, and will be useful to a variety of researchers (both theoretical and applied).